# Phosphoribosyl ubiquitination of SNARE proteins regulates autophagy during Legionella infection

Rukmini Mukherjee[1,2,3,9], Anshu Bhattacharya [1,2,9], Ines Tomaskovic[1], João Mello-Vieira [1,2], Melinda Elaine Brunstein[1], Marion Başoğlu[4], Tineke Veenendaal[5], Henry Bailey[1,2], Thomas Colby[6], Mohit Misra [1,2], Stefan Eimer[4], Judith Klumperman [5], Christian Münch [1], Ivan Matic[6,7] & Ivan Dikic [1,2,3,8 ✉]

## Abstract

***Legionella pneumophila*** is an intracellular pathogen that causes Legionnaires' disease. The bacteria release effector proteins, some of which remodel host autophagic-lysosomal pathways. One such effector is RavZ, which delipidates ATG8 proteins, making compromising autophagy in Legionella-infected cells. Here we show that SidE effectors also affect these pathways, by mediating phosphoribosyl-ubiquitination (PR-Ub) of the autophagic SNARE proteins STX17 and SNAP29. STX17 modification induces recruitment of STX17-positive membranes from the endoplasmic reticulum to Legionella-containing phagosomes, forming replicative vacuoles. Using proximity labeling, biochemistry and Legionella infection studies, we define a mechanism by which autophagy is hijacked by bacteria to recruit ER membranes to the bacterial vacuole, via a structure bearing autophagy markers but not fusing with lysosomes. Mass-spectrometric identification of PR-Ub sites and mutational studies show that phosphoribosyl-ubiquitination of STX17 alters its interaction with ATG14L, which causes ER membranes to be recruited to the bacterial vacuole in a PI3K-dependent manner. On the other hand, phosphoribosyl-ubiquitination of SNAP29 inhibits the formation of the autophagosomal SNARE complex (STX17-SNAP29-VAMP8) via steric hindrance, thus preventing the fusion of bacterial vacuoles with lysosomes.

**Keywords** Autophagy; Syntaxin17; Ubiquitin; *Legionella pneumophila*; Xenophagy
**Subject Categories** Autophagy & Cell Death; Membranes & Trafficking; Microbiology, Virology & Host Pathogen Interaction

## Introduction

Autophagy is a highly conserved mechanism in which long-lived proteins and damaged organelles are sequestered into double-membrane-bound vesicles called autophagosomes, which then fuse with lysosomes, resulting in the degradation of autophagic cargo (Dikic and Elazar, 2018; Fleming et al, 2022). Nutrient deprivation can trigger bulk autophagy, a generally non-selective process that recycles biochemical building blocks (Aman et al, 2021). In contrast, selective autophagy targets damaged organelles and intracellular pathogens via cargo receptors that mark the site of damage, carry LIR motifs that bind LC3, and can link cargos to the autophagic machinery (Lamark and Johansen, 2021; Gubas and Dikic, 2022). Intracellular bacteria trigger cell-autonomous immunity in the form of pathogen-specific selective autophagy (xenophagy) in an attempt to control bacterial replication and restore cellular homeostasis (Klionsky et al, 2021).

Autophagy involves the coordinated action of ATG (autophagy-related gene) proteins. Initially, the uncoordinated-51-like kinase (ULK1) complex comprising ULK1, ATG13 and FIP200, assembles at the phagophore assembly site. The class III PtdIns3K complex, consisting of Beclin1, ATG14L and phosphoinositide 3-kinase regulatory subunit 4 (PIK3R4) is then recruited to initiate phagophore membrane nucleation. PtdIns3P formed at this site interact with WD-repeat domain phosphoinositide-interacting proteins (WIPI1 and WIPI2), which in turn interact with the ATG proteins needed for autophagosome maturation. The induction of autophagy triggers the lipidation of LC3-I to form LC3-II, which is conjugated to phosphatidylethanolamine in the outer and inner phagophore membranes. ATG5, ATG12 and ATG16L form a complex that targets LC3-I to its membrane site of lipid conjugation. Mature autophagosomes then fuse with lysosomes via the STX17-SNAP29-VAMP8 complex, which corresponds to the autophagosomal soluble *N*-ethylmaleimide sensitive factor attachment protein receptor (SNARE) machinery.

[1]Institute of Biochemistry II, Faculty of Medicine, Goethe University, Frankfurt, Germany. [2]Buchmann Institute for Molecular Life Sciences, Goethe University, Frankfurt, Germany. [3]Max Planck Institute of Biophysics, Frankfurt, Germany. [4]Institute for Cell Biology and Neuroscience, Faculty of Biosciences, Goethe-University, Frankfurt, Germany. [5]Section Cell Biology, Center for Molecular Medicine, University Medical Center Utrecht, Utrecht University, Utrecht, The Netherlands. [6]Max Planck Institute for Biology of Ageing, Joseph-Stelzmann-Str. 9b, 50931 Cologne, Germany. [7]CECAD Cluster of Excellence, University of Cologne, Joseph-Stelzmann-Str. 26, 50931 Cologne, Germany. [8]Fraunhofer Institute for Translational Medicine and Pharmacology (ITMP), Frankfurt, Germany. [9]These authors contributed equally: Rukmini Mukherjee, Anshu Bhattacharya. ✉E-mail: dikic@biochem2.uni-frankfurt.de

Bacteria have evolved sophisticated mechanisms to hijack host cell autophagy to ensure bacterial survival and proliferation. The intracellular pathogen *Legionella pneumophila* is phagocytosed by macrophages, and phagosomes containing bacteria then recruit intracellular membranes to build replicative vacuoles known as *Legionella*-containing vacuoles (LCVs) that do not fuse with lysosomes. The formation of conventional ATG8+ autophagosomes is inhibited by the bacterial effector protein RavZ, which irreversibly delipidates ATG8 proteins on autophagosomal membranes (Choy et al, 2012; Yang et al, 2017). Furthermore, the *L. pneumophila* effector LpSPL inhibits basal autophagy by consuming host sphingolipids, which are needed for autophagosome biogenesis. Interestingly, strains lacking both RavZ and LpSPL are still not targeted by autophagy, indicating the presence of further bacterial mechanisms that help to evade lysosomal degradation (Rolando et al, 2016; Omotade and Roy, 2020).

Phosphoribosyl-linked serine ubiquitination (PR-Ub) is a non-canonical form of ubiquitination catalyzed by *Legionella* SidE effector proteins. This is unique to this bacterium and is different from serine ubiquitination noted in mammalian cells, where ubiquitin is conjugated to serine residues of substrates using a hydroxyester linkage (McClellan et al, 2019). The SidE class of effectors (SdeA, SdeB, SdeC and SidE) modify host ubiquitin by phosphoribosylation. The phosphoribosylated ubiquitin is then transferred to the serine residues of substrate proteins by a phosphodiester linkage by the same set of enzymes (Bhogaraju et al, 2016; Kotewicz et al, 2017). SdeA is structurally and mechanistically well-characterized (Akturk et al, 2018; Dong et al, 2018; Kalayil et al, 2018). PR-Ub levels in infected cells are regulated by the deubiquitinases DupA and DupB, which are specific for PR-Ub (Shin et al, 2020). SdeA activity is fine-tuned by the *Legionella* glutamylase SidJ, which inhibits SdeA catalytic activity (Bhogaraju et al, 2019; Black et al, 2019; Gan et al, 2019). Previous studies have linked PR-Ub to the fragmentation of the Rtn4-labeled tubular endoplasmic reticulum (ER), and to altered morphology of the Golgi body in *Legionella*-infected cells (Qiu et al, 2016; Kotewicz et al, 2017; Liu et al, 2021).

Proteomic analysis has shown that the SNARE proteins Syntaxin17 (STX17) and SNAP29 are PR-ubiquitinated during infection (Shin et al, 2020). STX17 is a part of the autophagosomal SNARE complex essential for autophagosome–lysosome fusion. The autophagosomal SNARE complex comprises a target (t)-SNARE present on the autophagosome (STX17), the cytosolic SNARE (SNAP29), and the vesicular (v)-SNARE present on the lysosome (VAMP8). The Qa-SNARE domain of STX17 interacts with the R-SNARE domain of VAMP8, and they are bonded by the Qb-SNARE and Qc-SNARE domains of SNAP29 to form a four-helix bundle that mediates the fusion of autophagosomal and lysosomal membranes (Itakura et al, 2012; Li et al, 2020). ATG14/Barkor/Atg14L, an essential component of the class III phosphatidylinositol 3-kinase complex (PtdIns3K) binds to the STX17-SNAP29 binary complex on autophagosomes and promotes SNARE-mediated autophagosome fusion with lysosomes (Diao et al, 2015; Liu et al, 2015). Interestingly, STX17+ autophagosomes are also formed in ATG conjugation-deficient cells, but these autophagosomes have a prolonged lifetime and the inner autophagosomal membrane breaks down more slowly during autophagosome–lysosome fusion (Tsuboyama et al, 2016).

STX17 also has an important role in autophagosome biogenesis. Under basal conditions, STX17 is present on the ER, mitochondria, at mitochondria-associated membranes (MAM) and in the cytosol.

It is unclear from which pool STX17 is recruited to autophagosomes. A study by Kumar et al showed cytosolic STX17 to interact with ATG8 proteins and IRGM on the autophagosome (Kumar et al, 2018). More recent reports showed that following the induction of autophagy, STX17 is phosphorylated by TBK1 at Ser202, which facilitates the assembly of an STX17+ pre-autophagosomal structure from the *cis*-Golgi (Kumar et al, 2019). These Golgi-derived vesicles then fuse with endosomal ATG16L1 membranes to form a hybrid pre-autophagosomal structure (HyPAS) which are the source of membranes for biogenesis of autophagosomes (Kumar et al, 2021). STX17 is also present at ER mitochondria contact sites where it interacts with ATG14L to initiate formation of cup-shaped structures called omegasomes, which supply phosphoinositol 3 phosphate positive membranes to initiate autophagosome biogenesis (Hayashi-Nishino et al, 2009; Hamasaki et al, 2013). A serine protease from Legionella called lpg1137 has been reported to modulate ER–mitochondria contacts by cleaving STX17, which reduces interaction with its interactor ATG14L (Arasaki and Tagaya, 2017).

In this study, we show that during the first 2 h of Legionella infection STX17+ membranes derived from the ER are recruited to bacterial phagosomes causing its maturation into hybrid vacuoles bearing markers of autophagosomes and endosomes. These compartments are ATG8-deficient and serve as replicative vacuoles for the intracellular pathogen in the early phase of infection. PR-Ub of autophagosomal SNARE proteins STX17 and SNAP29 is necessary for the formation and maintenance of these vacuoles. Identification of the sites of PR-Ub modification revealed that PR-Ub of STX17 in the Qa-SNARE domain (on S202) enhances its interaction with ATG14L, increasing autophagosome biogenesis from ER membranes. This mechanism is utilized by Legionella to recruit ER membranes to the early-stage bacterial phagosome. PR-Ub of SNAP29 in the Qb-SNARE domain (at S63) blocks its interaction with STX17, resulting in the formation of STX17+ LCVs that cannot fuse with the lysosome.

## Results

### *Legionella pneumophila* effectors SidE and RavZ are both necessary to block host xenophagy during infection

The *L. pneumophila* effector RavZ delipidates ATG8 proteins to block autophagosome formation within infected cells, making them phenotypically similar to ATG8 conjugation-deficient cells. Moreover, the autophagosomal SNARE proteins STX17 and SNAP29, which are required for autophagosome–lysosome fusion, were identified as putative substrates of PR-linked serine ubiquitination (Shin et al, 2020), suggesting a role of Legionella SidE effectors in the regulation of host cell autophagy (Figs. 1A and EV1A). As previously reported (Choy et al, 2012), LC3-II levels were lower in cells infected with wild-type (WT) bacteria compared to uninfected cells and those infected with a ΔR (RavZ-deficient) strain. The inhibition of LC3-II formation was more apparent when autophagic turnover was prevented by treating cells with bafilomycin A1. LC3-II formation was unaffected in cells that lacked all members of the SidE class of effectors (ΔS and ΔRΔS) and was hence devoid of any PR-Ub (Fig. EV1B). Next, endogenous LC3 was immunostained in infected A549 cells. Cells infected with RavZ-deficient

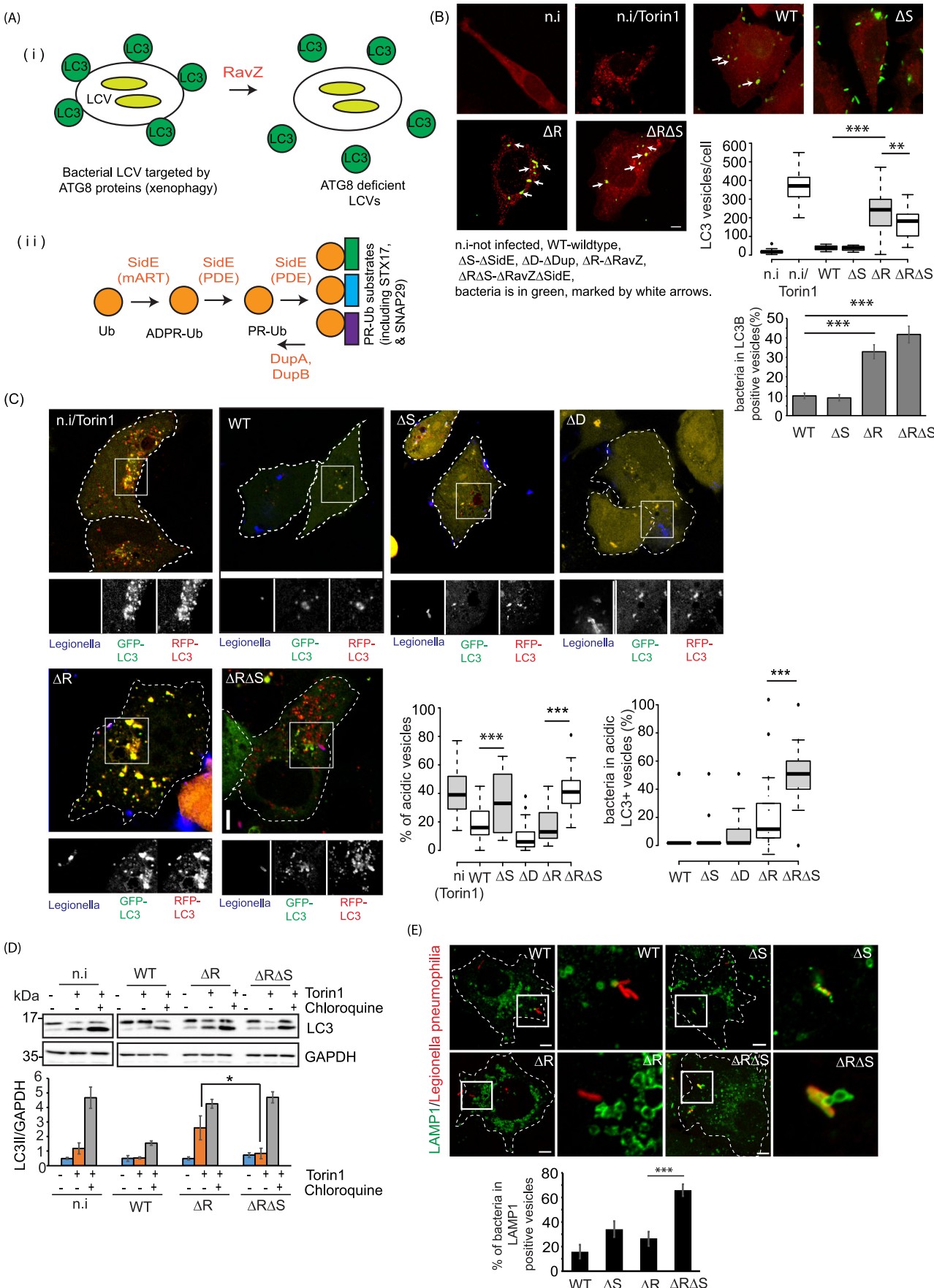

**Figure 1.   The *Legionella* effectors SidE and RavZ regulate autophagy during infection.**

(A) *Legionella* effectors that regulate autophagy. (B) A549 cells were infected with *Legionella* strains for 6 h, then endogenous LC3 was immunostained for confocal imaging. Uninfected cells were treated with 100 nM Torin-1 for 4 h to induce autophagosome formation. In the box plot, center lines show the medians; box limits indicate the 25th and 75th percentiles as determined by R software; whiskers extend 1.5 times the interquartile range from the 25th and 75th percentiles, outliers are represented by dots. $n = 34, 31$ cells taken from three independent experiments. $P$ value was calculated using two-tailed, type 3 Student's $t$ test, For the graph showing LC3 vesicles/cells: ***$P = 1.139E-11$ (WT vs $\Delta R$), **$P = 0.00853$ ($\Delta R$ vs $\Delta R\Delta S$). For the graph showing % of bacteria in LC3B-positive vesicles: ***$P = 1.5E-6$ (WT vs $\Delta R$), **$P = 1.74E-4$ ($\Delta R$ vs $\Delta R\Delta S$). Scale bar: 5 µm. (C) A549 cells were transfected with RFP-GFP-LC3 before infection with *Legionella* strains in the presence of 100 nM Torin-1 for 4 h. The cells were then fixed for immunostaining with a *Legionella*-specific antibody and confocal imaging. We counted the total number of puncta and the number of red puncta per cell in FIJI. In the box plots, center lines show the medians; box limits indicate the 25th and 75th percentiles as determined by R software; whiskers extend 1.5 times the interquartile range from the 25th and 75th percentiles. $n > 50$ cells taken from three independent experiments. $P$ value was calculated using two-tailed, type 3 Student's $t$ test, For the graph showing % acidic vesicles: ***$P = 6.27E-12$ (WT vs $\Delta S$), ***$P = 5.66E-31$ ($\Delta R$ vs $\Delta R\Delta S$). For the graph showing % of bacteria in acidic LC3B vesicles: ***$P = 1.98E-11$ ($\Delta R$ vs $\Delta R\Delta S$). Scale bar: 5 µm. Dotted lines indicate cell outlines drawn from thresholding images in FIJI. (D) A549 cells were infected with *Legionella* strains for 4 h in the presence of 200 nM Torin-1 and/or 100 µM chloroquine as indicated. The cells were lysed and analyzed by western blot with antibodies against LC3 and GAPDH. The data are means ± SEM of chemiluminescent intensity detected from three western blots taken from three independent experiments. $P$ value was calculated using a two-tailed, type 3 Student's $t$ test. *$P = 0.0216$. (E) A549 cells were infected with different strains of *Legionella* for 4 h before fixation and immunostaining with a LAMP1-specific antibody. We counted the number of LAMP1$^+$ bacteria per cell. The data are means ± SEM of 30 cells from three independent experiments. $P$ value was calculated using a two-tailed, type 3 Student's $t$ test. ***$P = 1.96E-6$. Scale bar: 5 µm. Dotted lines indicate cell outlines drawn from thresholding images in FIJI. ni not infected, WT wild-type, *Legionella*, $\Delta S$-$\Delta SidE$, $\Delta D$-$\Delta Dup$, $\Delta R$-$\Delta RavZ$, $\Delta R\Delta S$-$\Delta RavZ\Delta SidE$ *Legionella*). Source data are available online for this figure.

bacteria ($\Delta R$ and $\Delta R\Delta S$) produced a significantly greater number of autophagosomes; some of which encapsulated intracellular bacteria. $\Delta R$ had a significantly higher number of autophagosomes compared to $\Delta R\Delta S$. (Fig. 1B), suggesting that PR-Ub may regulate autophagic flux by modulating the clearance of autophagosomes. To determine the effect of SdeA on autophagy, we measured autophagic flux in HEK 293T cells expressing GFP-tagged SdeA or its mART mutant (E860AE862A). Cells were treated with the mTORC1 inhibitor Torin-1 for different periods of time, followed by western blotting to detect LC3-II. SdeA-expressing cells converted LC3-I to LC3-II more slowly than cells expressing catalytically deficient SdeA or those transfected with the control vector (Fig. EV1C). To determine the effect of PR-Ub on autophagic flux within infected cells, we visualized the transition from autophagosomes to autolysosomes by transfecting A549 cells with the tandem construct mRFP-GFP-LC3 prior to infection. Autophagosomes emit GFP and RFP fluorescence and thus appear as yellow puncta, whereas acidic autolysosomes quench GFP fluorescence and thus appear as red puncta. The induction of autophagy increases the quantity of yellow and red puncta, whereas late inhibition of autophagy (maturation of autophagosome or fusion with lysosome) increases the number of yellow puncta while reducing the number of red puncta. A549 cells treated with Torin-1 contained both red and yellow puncta before infection. Cells infected with WT *Legionella* contained very few LC3$^+$ puncta due to the activity of RavZ, but the $\Delta R$-infected contained a larger number of yellow autophagosomes that were not acidified to form red autolysosomes. The cells infected with $\Delta R\Delta S$ contained a greater proportion of red puncta and a larger number of LC3$^+$ bacteria compared to $\Delta R$-infected cells (Fig. 1C). Similarly, transient overexpression of SdeA and mRFP-GFP-LC3 also reduced the abundance of acidic red vesicles compared to the SdeA catalytic mutant E860AE862A (Fig. EV1D). Thus, suggested that RavZ and SidE may cooperate to shield bacteria from autophagy, the former preventing conjugating LC3 to autophagosomes and the latter potentially preventing autophagosome–lysosome fusion. We therefore monitored autophagosomal turnover in infected cells treated with combinations of Torin-1 (which induces autophagy) and chloroquine (which blocks lysosomal degradation). The difference in the amount of LC3-II between samples with and without

chloroquine represents the level of autophagic flux. LC3-II levels were higher in $\Delta R$-infected cells treated with Torin-1 than in $\Delta R\Delta S$-infected cells, indicating that SidE modulates the turnover of autophagosomes (Fig. 1D). Moreover, to check whether these early LC3$^+$ bacteria fuse with the lysosome, we measured the recruitment of LAMP1 to the different *Legionella* strains. Infection with the $\Delta S$ strain did cause some colocalization between LAMP1 and the bacteria, but it was much less than in $\Delta R\Delta S$, where most intracellular bacteria are completely engulfed in LAMP1$^+$ lysosomes (Fig. 1E). We also tested the intracellular replication of these strains in A549 cells and RAW264.7 macrophages. In both these cell lines, the single deletion strains $\Delta S$ and $\Delta R$ replicated less efficiently than WT controls, whereas the $\Delta R\Delta S$ strain showed the lowest replicative potential (Fig. EV1E). These results indicated that SidE and RavZ are both important to prevent the xenophagic targeting of bacteria.

## STX17 and SNAP29 are modified by PR-ubiquitination

We next focused on the role of the autophagosomal SNARE proteins STX17 and SNAP29, which were reported to be putative PR-ubiquitination substrates for SidE proteins (Shin et al, 2020). First, we confirmed the PR-ubiquitination of STX17 in infected cells. The PR-ubiquitination of FLAG-STX17 and V5-SNAP29 was observed by immunoblotting after enriching for PR-Ub substrates with a GST-DupA (H67A) trapping matrix using lysates from HEK 293T cells infected with Legionella for 2 h (Fig. 2A,B). Furthermore, the expression of wild-type SdeA (but not the inactive mART mutant) resulted in the appearance of PR-ubiquitinated STX17 and SNAP29 (Fig. EV2A,C). This PR-Ub modification of the SNARE proteins was lost when lysates were treated in vitro with WT DupA (Fig. EV2B,C).

We identified the serine residues on STX17 and SNAP29 modified by SdeA in an in vitro ubiquitination assay using GST-tagged STX17 and SNAP29 purified from *Escherichia coli* and incubated with SdeA, ubiquitin and NAD$^+$ (Fig. EV2D,E). The products of the in vitro PR-Ub assay were analyzed by high-resolution electron-transfer dissociation mass spectrometry (ETD-MS), revealing two PR-Ub sites on STX17 (S202 and S209) and one PR-Ub site on SNAP29 (S63) (Figs. EV3 and EV4A). The PR-Ub sites of STX17 are located within the Qa-SNARE domain, while SNAP29 is modified within the Qb-

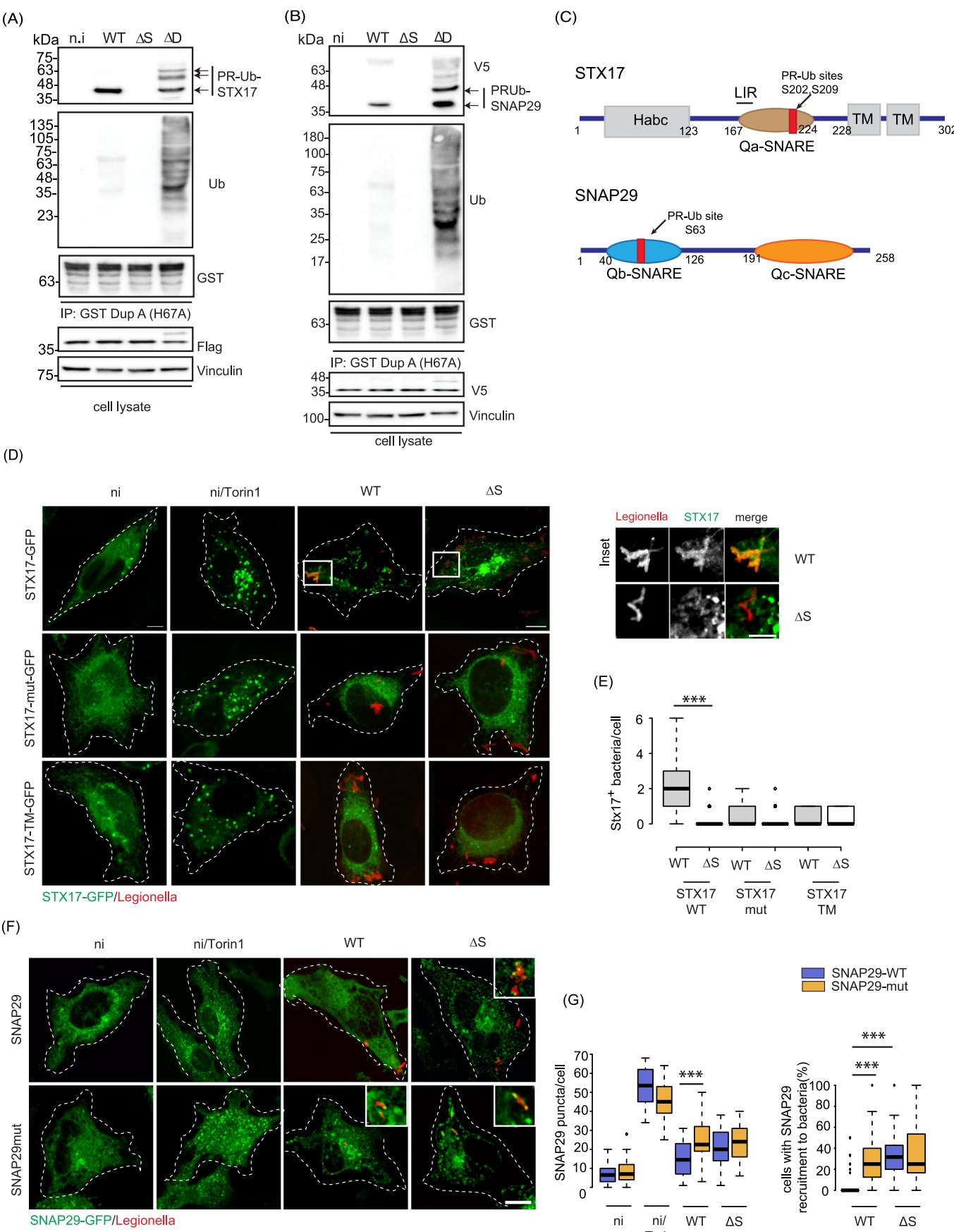

**Figure 2. STX17 and SNAP29 are PR-Ub modified during Legionella infection.**

(A) HEK 293T cells expressing FLAG-STX17 were infected with different strains of *Legionella* for 2 h. Lysates were used for GST pulldown with the DupA trapping mutant GST-DupA (H67A), followed by western blots with antibodies against FLAG, GST, and ubiquitin. Whole-cell lysates were blotted with antibodies against FLAG and vinculin as a loading control. This experiment was repeated three times with similar results. (B) HEK 293T cells expressing V5-SNAP29 were infected with different *Legionella* strains for 2 h. Lysates were used for GST pulldown with the DupA trapping mutant GST-DupA (H67A) followed by western blotting with antibodies against V5, GST, and ubiquitin. Whole-cell lysates were probed with antibodies against V5 and vinculin as a loading control. This experiment was repeated three times with similar results. (C) Domain architecture of STX17 and SNAP29 showing the PR-ubiquitination sites. (D) A549 cells expressing GFP-tagged STX17, STX17TM or the PR-Ub deficient mutant of STX17 were infected with *Legionella* strains for 2 h before fixation and immunostaining with a *Legionella*-specific antibody for analysis by confocal microscopy. Control cells were treated with 300 nM Torin-1 for 4 h. Dotted lines indicate cell outlines drawn from thresholding images in FIJI. Scale bar: 5 µm. Scale bar in inset: 2 µm. (E) The number of STX17$^+$ bacteria per cell were counted for ~50 cells taken three different experiments. In the box plots, center lines show the medians; box limits indicate the 25th and 75th percentiles as determined by R software; whiskers extend 1.5 times the interquartile range from the 25th and 75th percentiles, outliers are represented by dots. $N = 52$, 56 cells taken from three experimental replicates. $P$ value was calculated using two-tailed, type 3 Student's $t$ test, ***$P = 7.43E-8$. Scale bar: 5 µm. (F) The formation of WT and PR-Ub-deficient SNAP29-GFP puncta was monitored in *Legionella*-infected cells 4 h post-infection. Dotted lines indicate cell outlines drawn from thresholding images in FIJI. (G) SNAP29 puncta were counted in 50-µm$^2$ regions of interest using FIJI. In the box plot, center lines show the medians; box limits indicate the 25th and 75th percentiles as determined by R software; whiskers extend 1.5 times the interquartile range from the 25th and 75th percentiles. $n > 50$ cells taken from three independent experiments. $P$ value was calculated using two-tailed, type 3 Student's $t$ test, For the graph SNAP29 puncta/cell: ***$P = 6.82E-20$. For the graph % cells with SNAP29 recruitment to bacteria: ***$P = 2.27E-5$ (WT, SNAP29WT vs ΔS, SNAP29WT) ***$P = 2.56E-6$ (WT, SNAP29WT vs WT, SNAP29mut. Scale bar: 5 µm. Source data are available online for this figure.

SNARE domain (Fig. 2C). Mutation of three serine residues to alanine (STX17S195AS202AS209A) was necessary and sufficient to greatly reduce the PR-ubiquitination of STX17 in cells infected with WT or ΔS *Legionella* strains (Figs. EV2D and EV4B). Similarly, SNAP29 mutant (SNAP29S61AS63AS70A) was PR-Ub deficient (Fig. EV2E).

## Phosphoribosyl-ubiquitination facilitates the recruitment of STX17 to bacterial vacuoles

We investigated the formation of STX17$^+$ autophagosomes in A549 cells infected for 2 h WT and ΔS *Legionella*. STX17 was predominantly distributed in the ER of uninfected cells. However, the induction of autophagy by adding Torin-1 caused STX17 to form autophagosomal puncta (Fig. 2D). Infection with WT *Legionella* led to the formation of STX17-GFP vesicles; the number of vesicles was significantly lower in cells infected with the ΔS strain, showing PR-Ub was important for the formation of STX17$^+$ vesicles. The formation of STX17 vesicles was greatly reduced in cells expressing the STX17-GFP serine mutant (S195AS202AS209A) and in cells expressing a mutant of STX17 which only has the transmembrane hairpins of STX17 (STX17TM) (Itakura et al, 2012). Both the serine mutant and the STX17TM mutant can form autophagosomal puncta upon induction of autophagy with Torin1 treatment but are PR-Ub-deficient and hence cannot form STX17 vesicles during *Legionella* infection. The lack of PR-ubiquitination also blocked the recruitment of mutant STX17 to bacterial vacuoles (Fig. 2D,E). These results show that phosphoribosyl-ubiquitination of STX17 facilitates the formation of STX17 vesicles which are recruited to the early bacterial vacuole (2 h post-infection).

## STX17 is recruited to bacterial vacuoles from the ER in a PI3K-dependent manner

STX17 translocates from the ER to the Golgi body in serum-starved cells, where it associates with FIP200 and ATG13 to form pre-autophagosomal structures (Kumar et al, 2019). To determine whether translocation of STX17 to the golgi is necessary for recruitment of STX17 to bacterial phagosomes in Legionella infection, we treated cells with Brefeldin A which blocks COP-I-mediated transport between the ER and golgi at the ER exit sites. Brefeldin A treatment had no effect on STX17 recruitment to bacteria (Fig. EV5A). The formation of STX17$^+$

autophagic vesicles and the recruitment of STX17 to bacteria were reduced in cells treated with the PI3K inhibitor wortmannin (Fig. EV5B). STX17-positive ER membranes were also observed in close proximity to the bacterial vacuole through immune-electron microscopy (Fig. EV5C). These results suggest that the PR-Ub of STX17 is important for the recruitment of STX17 from the ER in a PI3K-dependent manner.

## PR-ubiquitination of SNAP29 prevents its recruitment to bacterial vacuoles

Given that STX17 is a component of bacterial vacuoles, we investigated whether its SNARE partner (SNAP29) was also recruited to bacteria. SNAP29 is a cytosolic protein that is recruited to autophagosomes when autophagy is induced. SNAP29-GFP-positive autophagosomes were observed in Torin1-treated cells. SNAP29-GFP expressing cells infected with WT Legionella had fewer SNAP29 puncta when compared to cells infected with the ΔS strain. Also, infection with ΔS *Legionella* led to recruitment of SNAP29-GFP to intracellular bacteria. In cells expressing the PR-Ub deficient mutant of SNAP29, SNAP29 forms puncta upon both WT and ΔS *Legionella* infection; some of which colocalize with bacteria (Fig. 2F,G). STX17 recruitment to the bacterial vacuole is inhibited and SNAP29 recruitment is enhanced in cells infected with ΔRΔS *Legionella* (Fig. EV5D,E). Similarly, the formation of SNAP29-GFP puncta was inhibited in Torin-1-treated cells expressing SdeA compared to cells expressing the catalytic mutant of SdeA or a control vector. The PR-Ub-deficient mutant of SNAP29 (S61AS63AS70A) formed autophagosomal puncta in SdeA-expressing cells treated with Torin-1 (Fig. EV5F,G). Taken together, these experiments indicate that PR-Ub of SNAP29 inhibits its recruitment to autophagosomes and bacterial vacuoles.

## PR-Ub converts the bacterial phagosome into an autophagosome-like vacuole

Having already observed the recruitment of STX17 to bacteria, we sought to determine whether other autophagy markers were also recruited to the bacterial vacuole 2 h post-infection. The early endosome marker Rab5 was recruited to intracellular bacteria along

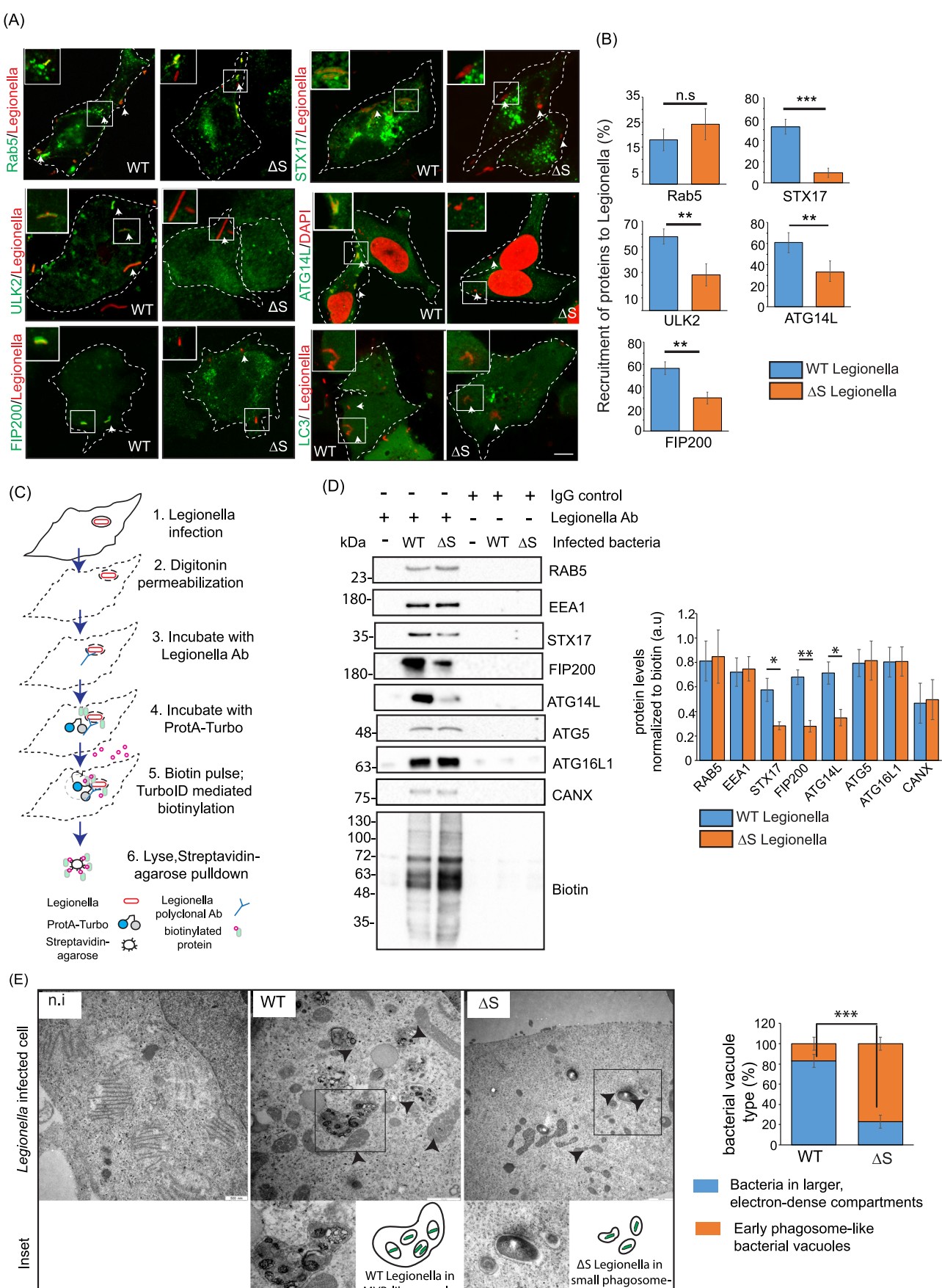

**Figure 3. Autophagy proteins are recruited to bacteria 1 h post-infection.**

(A) A549 cells were infected with WT or ΔS *Legionella* for 1 h, fixed and immunostained with the indicated antibodies to check for the recruitment of endocytic and autophagic markers to intracellular bacteria. Scale bar: 5 μm. White arrows mark intracellular bacteria. Dotted lines indicate cell outlines drawn from thresholding images in FIJI. (B) The experiment in (A) was quantified to measure the recruitment of each protein to intracellular bacteria. The total number of intracellular bacteria per cell and the number of bacteria which is surrounded by the protein marker were counted manually in FIJI to find the % of intracellular bacteria which were positive for the recruitment of the protein. The data are means ± SEM of 50 cells representing three experiments. *P* value was calculated using a two-tailed type 3 Student's *t* test. ***$P = 3.15E\text{-}5$ (STX17), **$P = 0.0066$ (ULK2), **$P = 0.01$ (ATG14L), **$P = 0.001$(FIP200), n.s. $P = 0.42$(Rab5). Dotted lines indicate cell outlines drawn from thresholding images in FIJI. (C) Proximity labeling of bacterial vacuoles in digitonin-permeabilized cells 2 h post-infection. (D) Western blots of the indicated proteins after streptavidin pulldown from lysates derived from cells treated with TurboID-ProtA and *Legionella* antibody. The data represent means ± SD taken from three independent experiments. *P* value was calculated using a two-tailed type 3 Students *t* test. n.s. $P = 0.44$ (Rab5), n.s. $P = 0.802$ (EEA1), *$P = 0.013$ (STX17), **$P = 0.006$ (FIP200), *$P = 0.03$(ATG14L), n.s. $P = 0.387$ (ATG5), n.s. $P = 0.523$ (ATG12), n.s. $P = 0.714$ (CANX). (E) HeLa cells expressing CD32 (for efficient uptake of Legionella) were uninfected (n.i.) infected with WT or ΔS Legionella for 6 h, fixed and prepared by following a protocol which was similar to that in (C), and imaged by TEM. In total, 30 images collected from three experiments were analyzed to count number of large electron-dense bacterial vacuoles and the number of early phagosome-like vacuoles. Error bars represent ± SD. *P* value was calculated using two-tailed type 3 Student's *t* test, ***$P = 1.008E\text{-}8$. Arrows mark intracellular bacteria in lysosome-like vesicles (in WT) or in smaller phagosomes (in ΔS) (WT wild-type, *Legionella*, ΔS-ΔSidE *Legionella*). Source data are available online for this figure.

with autophagosome initiation markers STX17, FIP200 and ULK2. However, the bacterial vacuole is LC3B-deficient due to the activity of the bacterial protease RavZ. The recruitment of these autophagy markers was PR-Ub dependent since cells infected with ΔS *Legionella* had significantly less recruitment (Fig. 3A,B). To confirm our immunofluorescence data, we used proximity labeling of the bacterial vacuole in cells fixed 2 h post-infection with WT or ΔS *Legionella*, permeabilized with 0.05% digitonin, and incubated with a *Legionella*-specific antibody to label the vacuole. We then added TurboID-tagged protein A, which binds the *Legionella* antibody and biotinylates proteins in the vicinity of the bacterial vacuole (Fig. 3C). Endosomal markers (Rab5 and EEA1) and proteins of the ATG8 conjugation machinery (ATG5, ATG16L) were found in the proximity of intracellular WT and ΔS *Legionella*. The recruitment of STX17, ATG14L, and FIP200 was facilitated by PR-Ub. Calnexin levels detected were low in both WT ΔS *Legionella* infection at 2 h.p.i. (Fig. 3D). To characterize the bacterial vacuoles further, we analyzed LCVs by electron microscopy. At 2 h.p.i., we observed endocytosed bacteria to be present in phagosomes which were small in size, contain a single bacterium per phagosome and were usually present near the cell periphery. At 6 h.p.i., these phagosomes mature into larger vacuoles which often contain more than one bacterium and have granular electron-dense material present in them. These compartments are probably representative of late compartments of the endocytic pathway (Fig. EV6A). At 6 h.p.i., ΔS Legionella were present in smaller phagosomes compared to the bacterial phagosomes observed in WT Legionella infection. WT Legionella-infected cells (at 6 h.p.i.) also had large lysosome-like compartments, which were not present in uninfected or ΔS Legionella-infected cells. These experiments highlight the role of PR-Ub in the maturation of the bacterial phagosome during the early phase of infection (Fig. 3E).

## The PR-ubiquitination of STX17 is important for the formation and maintenance of bacterial replicative vacuoles

To understand if STX17 was essential for the formation of these autophagosome-like vacuoles, we tested the recruitment of FIP200 and ATG14L to intracellular bacteria in cells treated with STX17 siRNA for 48 h. Knockdown of endogenous STX17 expression caused a significant decrease in recruitment of autophagy initiation markers to the bacterial vacuole (Fig. 4A). To check whether the

maturation and size of LCVs was dependent on STX17, we immunostained LCVs at a late stage of infection (12 h.p.i.). This showed that bacterial vacuoles were larger in cells infected with WT compared to ΔS *Legionella*, and in untreated cells compared to those treated with STX17 siRNA (Fig. 4B). We also performed an intracellular *Legionella* replication assay in A549 cells depleted of endogenous STX17 to determine the importance of STX17 for bacterial replication. The bacterial load decreased in cells treated with STX17 siRNA 48 h before infection (compared to mock siRNA treatment). The STX17 siRNA-dependent inhibition of intracellular bacterial replication was negligible in cells infected with the ΔS *Legionella* strains, indicating the process is dependent on PR-Ub (Fig. 4C). Furthermore, WT STX17 but not the PR-Ub-deficient mutant (S195AS202AS209A) was able to rescue the intracellular replication of WT bacteria in STX17-depleted cells (Fig. 4D).

## Proximity labeling of STX17 shows that STX17 interacts with components of the autophagic and endosomal pathways

To understand the role of STX17 in bacterial replication, we carried out STX17 proximity labeling experiments using infected cells. HeLa cells expressing doxycycline-inducible APEX2-FLAG-STX17 were infected with WT, ΔS, ΔR or ΔRΔS *Legionella* for 1 h. APEX2-STX17 was recruited to intracellular WT bacteria (Fig. EV6B) like the endogenous protein so we analyzed its biotin-labeled proteome by MS analysis (Fig. 4E). Samples were analyzed by 6-plex tandem mass tag (TMT) labeling, comparing (i) WT infection to uninfected cells (Fig. EV6D), (ii) WT to ΔS infection (Fig. EV6E), and (iii) ΔR to ΔRΔS infection (Fig. 4F). The biotin-labeled proteome of STX17 included proteins involved in the interconnected autophagic, secretory and endocytic pathways (Fig. 3B). Proteins that were preferentially enriched by infection with WT *Legionella* compared to uninfected cells included those related to autophagy (ATG3, ATG16L and BECN1), endocytosis (RAB5C, RAB8A, HGS1 and TFRC), ER-to-Golgi trafficking (ERGIC1, SEC23A, COPG1 and COPG2) and the Golgi and *trans*-Golgi network (STT3A, STT3B, LMAN2, B4GALT1, RAB34, ARF3 and ARF4). In uninfected cells, STX17 interacted with ER proteins, components of the protein translation machinery and ribosomal components (SIGMAR1, EIF2, SRP9, SRP14, RPL35, RPL23, and RPL14) (Fig. EV6D). Comparing the biotin-labeled-interaction landscape of infections with WT and ΔS *Legionella* strains, we observed the enrichment of proteins related to autophagosome formation (PIK3C3, PIK3C2, BECN, and ATG16L),

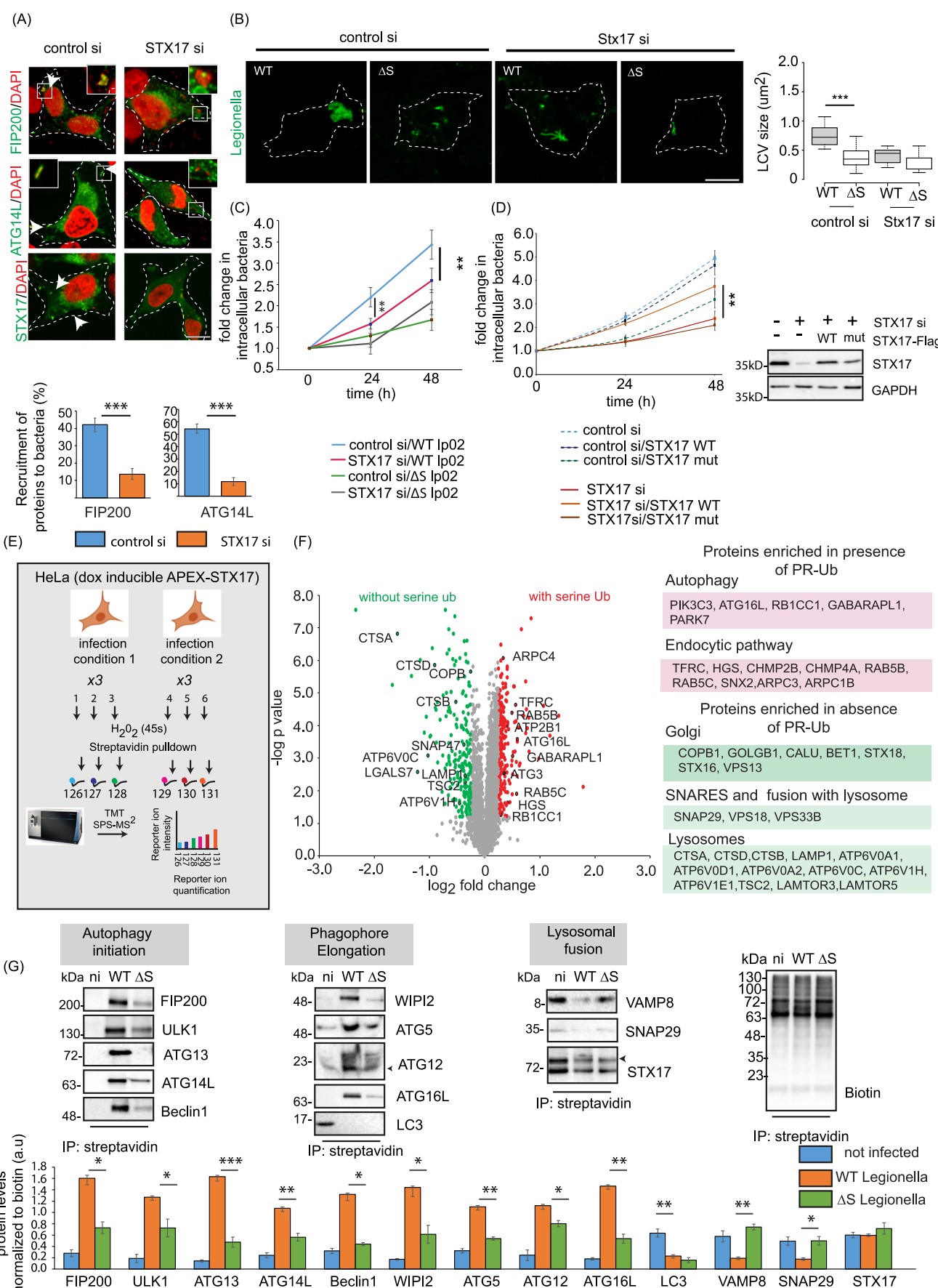

**Figure 4.  STX17 is crucial for the formation of a replicative vacuole and interacts with proteins of the endosomal and autophagy pathways during Legionella infection.**

(A) A549 cells were treated with control or STX17 siRNA for 48 h followed by infection with WT *Legionella*. Cells were fixed and stained for the indicated autophagy markers 1 h after infection. The data are means ± SEM of 50 cells representing three experiments. which were analyzed per sample to quantify recruitment of FIP200 and ATG14L to bacteria. *P* value was calculated using two-tailed type 3 Student's *t* test, ***$P$ = 5.83E-7 (FIP200), ***$P$ = 1.92E-12 (ATG14L), Scale bar: 5 µm. Dotted lines indicate cell outlines drawn from thresholding images in FIJI. (B) A549 cells were treated with STX17 or control siRNA for 48 h followed by infection with WT or ΔS *Legionella* for 12 h (MOI = 1). Cells were fixed for immunostaining with a *Legionella*-specific antibody followed by confocal microscopy. The LCV size was estimated in FIJI. In the box plot, center lines show the medians; box limits indicate the 25th and 75th percentiles as determined by R software; whiskers extend 1.5 times the interquartile range from the 25th and 75th percentiles, outliers are represented by dots. $n$ = 32, 31 cells taken from three independent experiments. p value was calculated using two-tailed, type 3 Student's *t* test, ***$P$ = 8.13E-16. Scale bar: 5 µm. Dotted lines indicate cell outlines drawn from thresholding images in FIJI. (C) A549 cells were treated with control or STX17 siRNA for 48 h followed by infection with WT or ΔS *Legionella*. Intracellular bacterial replication was assessed after 0, 24, and 48 h. Data are means ± SEM of three independent experiments. *P* value was calculated using two-tailed type 3 Student's *t* test, **$P$ = 0.00526 (WT, control vs STX17siRNA, 24 h), **$P$ = 0.00815 (WT, control vs STX17siRNA, 48 h). (D) A549 cells were treated with STX17 or control siRNA for 48 h followed by transfection with WT or PR-Ub-deficient STX17 for 24 h. Intracellular bacterial replication was assessed after 0, 24, and 48 h. Data are means ± SEM of three independent experiments **$P$ = 0.0077 (WT, STX17siRNA versus STX17 mutant, STX17siRNA, 48 h). Western blotting with STX17 antibody shows knockdown efficiency of STX17 siRNA and reconstitution with WT or mutant STX17. (E) Proximity labeling assay workflow. HeLa cells expressing doxycycline-inducible APEX-STX17 and CD32(for increasing the efficiency of *Legionella* uptake) were infected with *Legionella* for 2 h before treatment with biotin-tyramide and $H_2O_2$ followed by streptavidin pulldown. The samples were reduced, alkylated and digested with trypsin before MS analysis. Samples representing three biological replicates each of non-infected and *Legionella*-infected cells were analyzed in a single reaction by 6-plex TMT labeling. (F) Volcano plot showing changes in the biotin-labeled proteome following the infection of HeLa cells expressing APEX2-FLAG-STX17 with ΔR and ΔRΔS *Legionella* for 2 h, GO analysis of the biotin-labeled proteome showing pathways upregulated by infection with ΔR vs ΔRΔS *Legionella*. Data represents mean fold change of three experimental replicates per infection set ($n$ = 3). *P* value was calculated using two-tailed type 3 Student's *t* test and significant candidates were chosen having *P* value ≤ 0.01 and log2 (fold change) value minimum of ±0.5. Red and green indicate compartments containing proteins enriched following infection with ΔR and ΔRΔS *Legionella*, respectively. (G) Cells expressing doxycycline-inducible APEX-STX17 were infected with *Legionella* for 2 h before treatment with biotin-tyramide and $H_2O_2$ followed by streptavidin pulldown. The samples were analyzed by western blot with antibodies against proteins of the autophagic and endosomal pathways. The data represent means ± SD of three independent experiments. *P* value was calculated using two-tailed type 3 Students *t* test. WT vs ΔS *P* values: *$P$ = 0.01006 (FIP200), **$P$ = 0.0206 (ULK1), $P$ = ***0.0007 (ATG13) **$P$ = 0.005 (ATG14), *$P$ = 0.0111(Beclin1), *$P$ = 0.0219 (WIPI2), **$P$ = 0.0038 (ATG5), *$P$ = 0.018 (ATG12), **$P$ = 0.001 (ATG16), **$P$ = 0.0036 (VAMP8), *$P$ = 0.0424 (SNAP29). (n.i. not infected, WT wild-type, *Legionella*, ΔS ΔSidE *Legionella*). Source data are available online for this figure.

endocytosis (HGS, TFRC, AP2M1, AP2B1, and SNX6), and the *trans*-Golgi network (ARF6, ARF1, RAB10 and M6PR) in WT infections, whereas STX17 preferentially interacted with proteins of the ER-Golgi intermediate compartment (ERGIC) and Golgi body (SEC23B, SEC24C, SEC31A, ERGIC2, GOLGA2, GOLGA5, LMAN2 and LMAN1), SNARE proteins needed for vesicle fusion (SNAP29 and SNAP47), components of the HOPS complex needed for lysosomal fusion (VPS41, VPS18 and VPS11) and mitochondrial proteins (MAVS, MFN2 and TIMM21) in infections with the ΔS strain (Fig. EV6E). Western blots of these samples after streptavidin pulldown showed the proximity of STX17 to autophagy initiation factors (FIP200, ATG13, ULK1, ATG14L and Beclin1), the PI3P binding protein WIPI2, and the LC3 conjugation machinery (ATG5, ATG12 and ATG16L) in cells infected with WT *Legionella* but not in uninfected cells or those infected with ΔS strains. On the other hand, STX17 did not interact with autophagosomal SNARE components (VAMP8 and SNAP29) in cells infected with WT *Legionella* (Figs. 4G and EV6C).

Next, we compared the biotin-labeled proteome of cells expressing APEX2-STX17 and infected with *Legionella* strains ΔR or ΔRΔS. In the absence of RavZ, autophagy components such as GABARAPL1, ATG16L1, and FIP200 were enriched following infection with the ΔR strain, whereas STX17 interacted with several lysosomal proteins, including cathepsins (CTSB and CTSD), LAMP proteins, the mTOR complex (LAMTOR3, LAMTOR5 and TSC2) and the lysosomal V-ATPase (ATP6V0A1, ATP6V0A2, ATPV0C and ATP6V1H) following infection with the ΔRΔS strain (Fig. 4E,F).

## PR-ubiquitination of STX17 increases its interaction with ATG14L, while PR-Ub of SNAP29 inhibits the formation of the autophagosomal SNARE complex

SNAP29 was not recruited to autophagosomes in the presence of SidE so we determined whether its interaction with other components of the autophagosomal SNARE complex was affected by PR-Ub. Structural

rearrangements were recently characterized in the STX17-SNAP29-VAMP8 complex during autophagosome–lysosome fusion (Li et al, 2020). Based on this structure, it was evident that the PR-Ub site of SNAP29 is at the interface that binds STX17 and VAMP8. We added PR-Ub (PDB ID: 5M93) manually to the S63 residue of SNAP29 in the structure of the STX17–SNAP29–VAMP8 complex (7BV6) using Pymol, revealing that the modification is likely to sterically hinder the formation of the four-helix bundle that triggers the fusion of opposing membranes (Fig. 5A,B). To test this hypothesis, we modified pure SNAP29 protein in an in vitro reaction with purified SdeA and His-tagged Ubiquitin and enriched the PR-Ub modified SNAP29 through nickel-NTA-based affinity purification. Purified PR-Ub modified SNAP29 was incubated with HEK 293T lysate, followed by immunoprecipitation of SNAP29 and immunoblotting. This showed the PR-Ub SNAP29 interacted with less VAMP8 and STX17 from HEK 293T lysate. Its interaction with ATG14L was unperturbed by the PR-Ub modification. GST-tagged STX17 was also PR-Ub modified in vitro in a similar protocol as SNAP29, followed by enriching the PR-ubiquitinated form by nickel-NTA-based affinity chromatography. PR-Ub modified STX17 showed increased binding to ATG14L and SNAP29 in HEK 293T lysate. Other autophagy initiation factors: FIP200 and Beclin1 (which were recruited to bacterial vacuoles and enriched in the STX17 biotinylation landscape) did not interact with GST-tagged STX17 (Fig. 5C). Purified unmodified or PR-Ub modified SNAP29 was incubated with equimolar amounts of VAMP8, ATG14L and GST-STX17, followed by antibody-based immunoprecipitation of SNAP29. PR-Ub modified SNAP29 interacted with less STX17 and VAMP8 (Fig. 5D). Similarly, when PR-Ub modified STX17 was incubated with equimolar amounts of SNAP29, VAMP8 and ATG14, an increased interaction with SNAP29 and ATG14L was noted (Fig. 5E). From these experiments, we inferred that PR-Ub of STX17 in Qa- SNARE domain facilitates its interaction with SNAP29 and ATG14L both of which are known to interact with the Qa-SNARE domain of STX17. On the other hand, PR-Ub modification in the Qb-

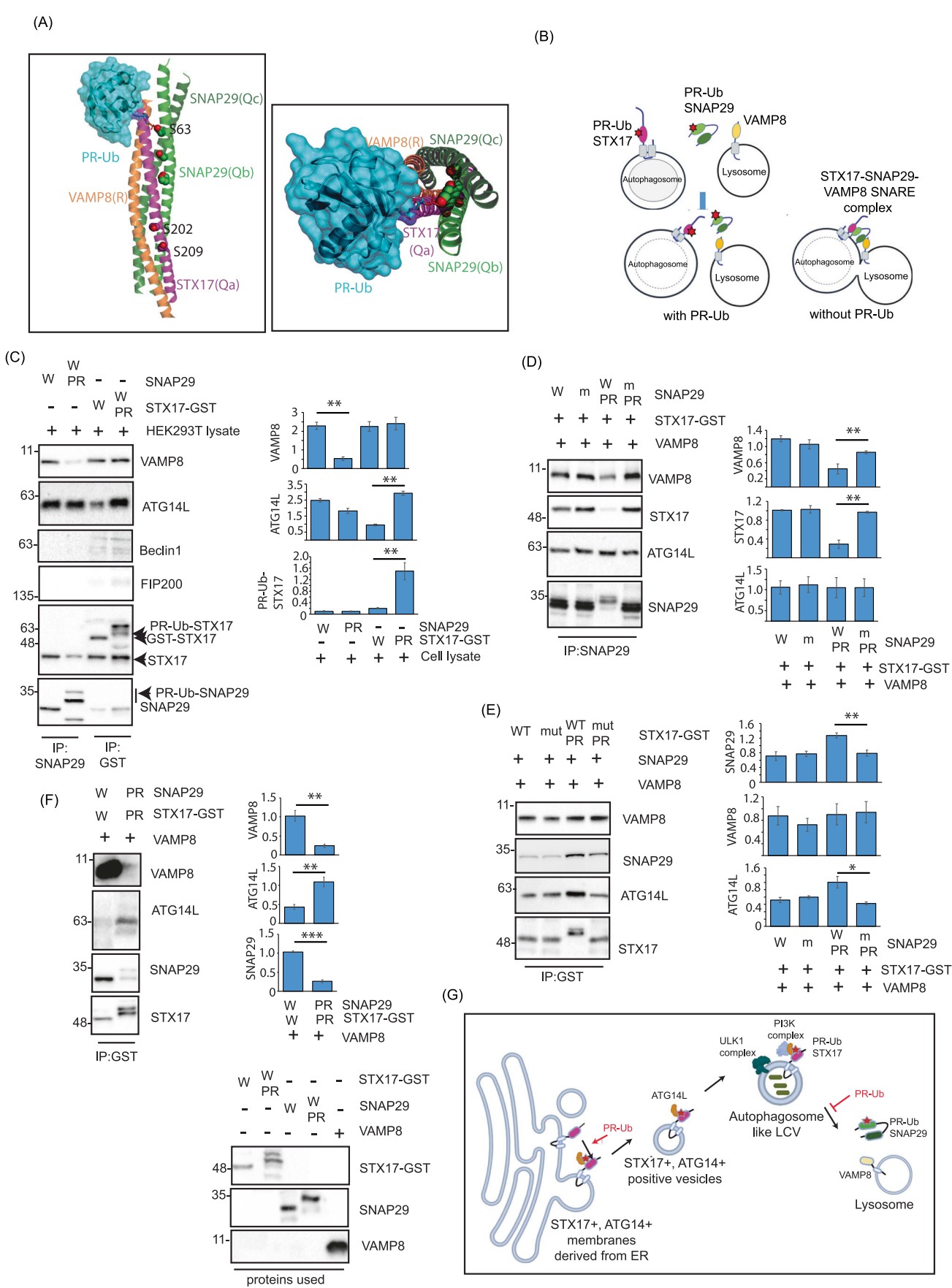

**Figure 5. PR-Ub of STX17 increases its interaction with ATG14L; while PR-Ub of SNAP29 blocks the mutual interaction between SNARE domains of STX17 and SNAP29.**

(A) PR-Ub (PDB ID: 5M93) was manually attached to residue S63 of SNAP29 in the structure of the STX17-SNAP29-VAMP8 complex (7BV6) using Pymol. (B) Schematic showing the formation of the autophagosomal SNARE complex, and the effect of PR-ub on its formation. (C) Untagged SNAP29 and STX17-GST was purified from *E. coli* and incubated with His-tagged Ub in an in vitro PR-ub assay. PR-Ub modified STX17 and SNAP29 were enriched by nickel-NTA-based affinity purification and incubated with 100 µg protein lysate from HEK 293T cells. This was followed by GST-pulldown of STX17 and antibody-based immunoprecipitation of SNAP29 to check for alterations in protein-protein interactions upon PR-Ub. The data represents means ± SD of three independent experiments *P* value was calculated using two-tailed type 3 Students *t* test 0.001 < **$P \le$ 0.01). **$P =$ 0.004 (VAMP8), **$P =$ 0.001 (ATG14L), **$P =$ 0.022 (PR-Ub STX17). (D) WT and PR-Ub-modified SNAP29 were incubated with equimolar amounts of purified GST-STX17, VAMP8 and ATG14L. Interactors of PR-Ub-modified SNAP29 were observed by immunoblotting. The data represent means ± SD of three independent experiments. *P* value was calculated using two-tailed type 3 Students *t* test. **$P =$ 0.0068 (STX17), **$P =$ 0.0059 (VAMP8). (E) WT or PR-Ub modified GST-STX17 were incubated with equimolar amounts of purified GST-STX17, VAMP8 and ATG14L. Interactors of PR-Ub modified STX17 were observed by immunoblotting. The data represent means ± SD of three independent experiments. *P* value was calculated using two-tailed type 3 Students *t* test. **$P =$ 0.042 (ATG14L), **$P =$ 0.029 (SNAP29). (F) WT or PR-Ub modified forms of both STX17 and SNAP29 were incubated with VAMP8 and ATG14L followed by immunoprecipitation of GST-STX17 and immunoblotting to check for protein-protein interactions. Pure proteins were run as the reaction inputs and immunoblotted with indicated antibodies. The data represent means ± SD of three independent experiments. *P* value was calculated using two-tailed type 3 Students *t* test. **$P =$ 0.0208 (ATG14L), **$P =$ 0.023 (VAMP8), ***$P =$ 0.00046 (SNAP29). (G) Schematic showing the effect of PR-Ub of STX17 and SNAP29 on bacterial vacuole formation. PR-Ub of STX17 enhances interactions with ATG14L which results in the recruitment of ER membranes to the bacterial vacuole.PR-Ub of SNAP29 prevents fusion of the STX17-positive vacuoles with the VAMP8-positive lysosomes. Source data are available online for this figure.

SNARE domain (on S63) of SNAP29 sterically inhibits its interaction with STX17, blocking the formation of the four-helix bundle of SNARE domains that is essential for the fusion of the autophagosome and lysosomal membranes (Fig. 5B).

In WT Legionella infection or in SdeA-expressing cells both STX17 and SNAP29 are modified by PR-Ub. When such conditions are recapitulated in vitro by incubating PR-Ub modified STX17 with PR-Ub modified SNAP29 and their interaction partners VAMP8 and ATG14L, PR-Ub modified STX17 shows increased interaction with ATG14L but significantly reduced the interaction with SNAP29 and VAMP8 (Fig. 5F). These results suggest that during Legionella infection, the PR-ubiquitination of STX17 in the Qa-SNARE domain increases its interaction with ATG14L which facilitates the recruitment of ER membranes to the bacterial phagosome in a PI3K-dependent manner. Presence of ATG14L on STX17-positive bacterial vacuoles does not promote the fusogenic activity of the SNARE domain due to the presence of PR-Ub modified SNAP29, which sterically inhibits the SNARE-dependent fusion of the autophagosome with the lysosome (Fig. 5G).

## Discussion

We have investigated the role of phosphoribosyl-ubiquitination in the formation of bacterial vacuoles during the early steps of *Legionella* infection while preventing the fusion of these vacuoles with lysosomes. During the first 10 min of infection, a sub-population of *Legionella* is present in Rab5$^+$ phagosomes, which lack late endosome markers and escape fusion with the lysosome (Ku et al, 2012; Gaspar and Machner, 2014). We found that early phagosomes recruit STX17$^+$ATG14L+ membranes from the ER in a PR-Ub dependent manner. This converts the bacterial vacuole into a hybrid compartment which has endosomal markers, and components of PI3K and ULK1 complex but is ATG8-deficient. These vacuoles resemble late endosomal compartments, appear electron-dense and contain several bacteria as seen in electron micrographs. The STX17+ vacuoles are protected from lysosomal degradation by the PR-Ub of SNAP29, which prevents their fusion with VAMP8+ lysosomes. Therefore, the absence of ATG8 lipidation (due to RavZ activity) and PR-ubiquitination of autophagosomal SNARE proteins prevent the fusion of bacterial vacuoles with the lysosome.

The PR-ubiquitination of STX17 is necessary for its recruitment to bacterial vacuoles, but it is unclear whether the PR-Ub-modified form of STX17 is solely present on the bacterial vacuole. It is also interesting to consider whether conventional ubiquitination regulates STX17 during autophagosome biogenesis and lysosomal fusion. The PR-ubiquitination of SNAP29 in the Qb-SNARE domain sterically hinders the formation of the four-helix bundle required to form the autophagosomal SNARE complex. SNAP29 is also implicated in several other vesicle trafficking events, including the regulation of secretory traffic from the Golgi body (Rapaport et al, 2018). SNAP29 activity is also regulated by post-translational modifications such as phosphorylation and the addition of *O*-linked β-*N*-acetylglucosamine (Smeele and Vaccari, 2023). A recent study by Jian et al shows how O-GlyNAcylation of SNAP29 reduces its participation in the SNARE complex and causes SNAP47 to replace it in the SNARE complex (Jian et al, 2024). Whether PR-Ub affects these non-canonical functions or modifications of SNAP29 will be interesting to study.

We can also speculate whether the generation of bacterial vacuoles is similar to non-canonical autophagic mechanisms such as LC3-associated phagocytosis (LAP) (Martinez et al, 2015). Electron microscopy revealed the presence of bacteria within single-membrane vesicles during the early-stage of infection. We did not observe significant recruitment of LC3 to WT bacteria at any stage of infection which rules out the possibility of LAP. Since the discovery of RavZ, *Legionella*-infected cells have been considered a mostly autophagy-deficient system due to the defect in LC3 conjugation. However, to maintain a cellular niche that is suitable for bacterial proliferation, the housekeeping functions of autophagy should remain operational. Low levels of autophagy may be needed to recycle macromolecules into nutrients necessary for bacterial survival. Basal levels of autophagy may reflect the use of STX17-mediated autophagy, which occurs in ATG-conjugation-deficient systems (Tsuboyama et al, 2016). In this case, the autophagosome is often elongated due to the inefficient degradation of the inner autophagosomal membrane (Tsuboyama et al, 2016). In our experiments using *Legionella*-infected cells, we observed elongated STX17-coated bacterial vacuoles surrounding individual rod-shaped bacteria 2 h post-infection.

During the initial stages of infection, the bacteria reside in PI(3)P$^+$ phagosomes that are soon converted to PI(4)P$^+$ vesicles (Weber

et al, 2014), which recruit ER markers such as calnexin. PR-ubiquitination of STX17 may be an important molecular mechanism by which ER membranes are recruited to the bacteria in the initial phase of Legionella infection. This STX17-dependent recruitment may be like the molecular changes that occur during autophagosome biogenesis at omegasomes. The later stages of vacuolar expansion are dependent on Atlastin-mediated membrane remodeling. This was shown in *Dictyostelium discoideum*, where the ER GTPase Atlastin3 (Sey1) was needed for ER remodeling around the vacuole to facilitate expansion (Steiner et al, 2017).

FAM134B and FAM134C were also PR-ubiquitinated by SidE during bacterial infection (Shin et al, 2020). These proteins have reticulon homology domains (which induce curvature in membranes) and LIR motifs (which interact with LC3) and can therefore generate ER-derived autophagosomes by ER-phagy (Hübner and Dikic, 2020). It is unclear whether this process, mediated by FAM134, contributes to the recruitment of membranes to bacteria in the absence of ATG8 conjugation. Our analysis of SidE effectors and their substrates indicates that PR-ubiquitination is an important mechanism utilized by *Legionella* to hijack intracellular membranes from the ER.

STX17 is a substrate of the *Legionella* serine protease Lpg1137, and although the levels of STX17 are unaffected by the protease 1 h post-infection, almost complete Lpg1137-dependent degradation has occurred after 4 h (Arasaki and Tagaya, 2017). Cleavage of STX17 by Lpg1137 eliminates STX17-ATG14L interactions at ER–mitochondria contact sites. We observed STX17⁺ vesicles and the recruitment of STX17 to bacteria during the first 2 h post-infection. At later time points, intracellular bacteria do not display STX17 or other autophagy markers but are instead enclosed in ER-like compartments. We speculate that the transition from bacterial vacuoles resembling autophagosomes to ER-derived LCVs may reflect the degradation of STX17 by Lpg1137.

During the initiation of autophagy, two major protein complexes are recruited to STX17-positive phagophore assembly sites: the PI(3)-kinase complex—which includes Vps34, Vps15, ATG14L, and Beclin1—and the ULK1 complex, composed of ATG101, ULK1/2, FIP200, and ATG13. Under autophagy-inducing conditions, the cellular kinase TBK1 phosphorylates STX17 at serine 202, a modification that promotes the recruitment of ATG13 and FIP200, facilitating the formation of the mammalian pre-autophagosomal structure (mPAS).

Interestingly, during *Legionella* infection, the pathogen exploits the same serine residue (S202) for phosphoribosyl-linked ubiquitination (PR-Ub). This modification enhances STX17's interaction with ATG14L, a component of the PI3K complex. Although phosphorylation and PR-ubiquitylation represent distinct post-translational modifications with potentially different impacts on STX17's interaction landscape, both ultimately converge on a similar functional outcome: promoting the formation of autophagosomes during canonical autophagy, or autophagosome-like bacterial vacuoles during *Legionella* infection.

The recruitment of STX17⁺ membranes to bacteria was unaffected by brefeldin A which block COPI-mediated transport to the golgi. These data suggest that STX17⁺ membrane recruitment occurs at the ER and these membranes may represent PI3P-positive omegasomes which are known to be important sites of autophagosome biogenesis (Hayashi-Nishino et al, 2009).

STX17 is required for the formation of HyPAS, which appears to be derived from FIP200⁺ vesicles originating from the *cis*-Golgi and ATG16L1⁺ endosomal membranes. The data from our immuno-fluorescence and proximity labeling studies of STX17 suggest that the bacterial vacuoles formed during the initial stages of infection are derived from hybrid phagosomes (which have early endosome markers like Rab5, EEA1, and ATG16L) and STX17⁺ pre-autophagosomes originating from the ER. The recruitment of ATG16L and other endosomal markers does not require PR-ubiquitination but PR-ubiquitination of STX17 is important to convert these endosome-like early phagosomes to bacterial vacuoles that have autophagy markers.

# Methods

## Reagents and tools table

| Reagent/resource | Reference or source | Identifier or catalog number |
|---|---|---|
| **Experimental models** | | |
| HEK 293T | ATCC | CRL-11268 |
| HeLa | ATCC | CRM-CCL-2 |
| A549 | ATCC | CRM-CCL-185 |
| Legionella pneumophila | Lab of Dr. Zhao-Qing Luo | |
| **Recombinant DNA** | | |
| FLAG-STX17 | Addgene | Addgene #45911 |
| pcDNA 3.2/V5-DEST SNAP29-V5 | Addgene | Addgene #69821 |
| ProtA-Turbo | Gift from Michiel Vermeulen | |
| STX17-ser mutant | Lab generated using STX17 coding sequence of STX17 from addgene #45911 | |
| SNAP29-ser mutant | Lab generated using SNAP29 coding sequence of SNAP29 from addgene #69821 | |
| **Antibodies** | | |
| STX17 | Proteintech | 17815-1-AP |
| GAPDH | Cell signaling technology | 2118 |
| GFP (for western blotting) | Santa Cruz Biotechnology | sc-9996 |
| GFP (for immune-electron microscopy) | Rockland | 600–106-215 |
| LC3 | Cell Signaling Technology | 2775 |
| VAMP8 | Cell Signaling Technology | 13060 |
| SNAP29 | Cell Signaling Technology | 3013 |
| RAB5 | Cell Signaling Technology | 3547 |
| RAB7 | Cell Signaling Technology | 9367 |
| ATG16L | abcam | ab188642 |
| ATG12 | Cell Signaling Technology | 4180 |
| Beclin1 | Cell Signaling Technology | 3738 |

| Reagent/resource | Reference or source | Identifier or catalog number |
|---|---|---|
| FIP200 | Proteintech | 17250-1AP |
| LAMP1 | Cell Signaling Technology | 9091 |
| *Legionella* | Abcam | 20943 |
| Ubiquitin | Cell Signaling Technology | 3933 |
| **Oligonucleotides and other sequence-based reagents** | | |
| STX17 siRNA | Santa Cruz Biotechnology | sc-92492 |
| SNAP29 siRNA | Santa Cruz Biotechnology | sc-76531 |
| **Chemicals, enzymes, and other reagents** | | |
| GFP trap beads | ChromoTek | gta-100 |
| biotin-tyramide | Sigma-Aldrich | SML2135 |
| Sodium azide | Sigma-Aldrich | S2002 |
| Trolox | EMD Millipore | 648471 |
| Hydrogen peroxide | Sigma-Aldrich | 1.93007 |
| Glutaraldehyde | Sigma-Aldrich | G5882 |
| DTT | Sigma-Aldrich | 646563 |
| Torin1 | Sigma-Aldrich | 475991 |
| Streptavidin-agarose | Cytiva | 17-5113-01 |
| **Software** | | |
| ImageJ/FIJI | https://imagej.net/ij/ | |
| Maxquant | https://www.maxquant.org/ | |
| Perseus | https://maxquant.net/perseus/ | |
| **Other** | | |

## Plasmid construction

FLAG-STX17 was a gift from Noboru Mizushima (Addgene #45911; RRID Addgene_45911), and pcDNA 3.2/V5-DEST SNAP29-V5 was a gift from Anne Brunet (Addgene #69821; RRID Addgene_69821). The coding sequence of STX17 from FLAG-STX17 was transferred to vectors pEGFPC1 and pGEX6P1 for expression in mammalian and bacterial cells, respectively. For proximity labeling, STX17 was tagged on the N terminus with FLAG-APEX2 in the doxycycline-inducible vector pcDNA5 by Gibson assembly. STX17 and SNAP29 serine mutants were generated by mutagenic PCR and site-directed mutagenesis. The ProtA-Turbo plasmid was a kind gift from Michiel Vermeulen.

## Protein purification

GST-tagged SNAP29 and STX17 (amino acids 1–224) and their PR-Ub-deficient serine mutants were purified from *E. coli* as previously described (Shin et al, 2020). Briefly, BL21(DE3) competent cells (NEB) were transformed with plasmids and grown in lysogeny broth at 37 °C to an $OD_{600}$ of 0.6–0.8. Protein expression was induced by adding 0.5 mM isopropyl-D-thiogalactopyranoside (IPTG) overnight at 18 °C. Lysates were incubated with glutathione Sepharose resin pre-equilibrated with lysis buffer (50 mM Tris-HCl (pH 7.5), 150 mM NaCl, 3 mM DTT), and non-specific proteins were cleared by washing

twice with wash buffer (50 mM Tris-HCl (pH 7.5), 300 mM NaCl, 3 mM DTT. Proteins were eluted in 50 mM Tris-HCl (pH 7.5), 150 mM NaCl, 15 mM glutathione and exchanged into storage buffer (20 mM Tris-HCl pH 7.5, 100 mM NaCl) before further use. Myc-ATG14L was purchased from abcam (ab225977), VAMP8 was purchased from Origene (TP500318).

## Antibodies and siRNA

Human STX17 siRNA was used from Santa Cruz Biotechnology (catalog no: sc-92492). We used the following antibodies and dilutions: STX17 (cat. no. 17815-1-AP, Proteintech; 1:1000), GAPDH (cat. no. 2118, Cell Signaling Technology; 1:2000), GFP trap beads (cat. no. gta-100, ChromoTek), GFP (cat. no. sc-9996, Santa Cruz Biotechnology; 1:2000), GFP for immune-electron microscopy (cat no. 600–106-215, Rockland), biotin for immuno-electron microscopy (cat. no. 100–4198, Rockland), LC3 (cat. no. 2775, Cell Signaling Technology; 1:2000), VAMP8 (cat. no. 13060, Cell Signaling Technology; 1:1000), SNAP29 (cat. no. 3013, Cell Signaling Technology; 1:2000), RAB5 (cat. no. 3547; 1:1,000), RAB7 (cat. no. 9367, Cell Signaling Technology; 1:2000), ATG16L (cat. no. ab188642, Cell Signaling Technology; 1:1000), ATG12 (cat. no. 4180, Cell Signaling Technology; 1:1000), Beclin1 (cat. no. 3738, Cell Signaling Technology; 1:1000), FIP200 (cat. no. 17250-1AP, Proteintech; 1:1000), LAMP1 (cat. no. 9091, Cell Signaling Technology; 1:2000), and *Legionella* (cat. no. 20943, Abcam; 1:4000), Ubiquitin (Cat. no: 3933, Cell Signaling Technology, 1:1000).

## Immunoprecipitation and western blotting

Cells were lysed in 50 mM Tris-HCl (pH 7.5) containing 150 mM NaCl and 1% Triton X-100. For the immunoprecipitation of STX17-GFP, lysates containing 1 mg protein were incubated with GFP beads in immunoprecipitation buffer (50 mM Tris-HCl pH 7.5, 150 mM NaCl, 0.5% Triton X-100), then washed (50 mM Tris-HCl pH 7.5, 300 mM NaCl, 0.5% Triton X-100) and boiled with SDS sample buffer. For western blotting, 20-µg samples were loaded onto 10% Tris-glycine gels and fractionated at 150 V for 1.5 h, followed by transfer to a PVDF membrane for 2 h at 300 mA and incubation with the antibodies listed above.

## GST pulldown assay

Cells were lysed (20 mM Tris-HCl pH 7.5, 150 mM NaCl, 1% Triton X-100) and precleared by incubating with 30 µL GST beads for 1 h to reduce non-specific binding. We then added 1 mg of the precleared lysate to 5 µg of GST-tagged protein and 20 µL glutathione Sepharose resin, and incubated at 4 °C for 2 h on a rotary shaker. The resin is then washed (20 mM Tris-HCl pH 7.5, 300 mM NaCl, 1% Triton X-100), boiled with SDS sample buffer and analyzed by western blot as described above.

## PR-Ub assay

In all, 5 mM GST-tagged STX17 and SNAP29 were incubated at 37 °C for 1 h with 25 mM of purified untagged ubiquitin, 1 mM $NAD^+$ and 1–2 mM SdeA in 50 mM Tris-HCl (pH 7.5) and 50 mM NaCl. The reaction mixture was processed as described for western blotting above, and probed with antibodies specific for ubiquitin and GST. PR-Ub-specific deubiquitination assays were performed

by incubating PR-Ub proteins with 1 μg of GST-DupA at 37 °C for 1 h in a buffer containing 50 mM Tris-HCl (pH 7.5) and 150 mM NaCl. For enrichment of PR-Ub modified STX17 and SNAP29, his-tagged ubiquitin was used instead of untagged ubiquitin, followed by enrichment of PR-Ub modified proteins by Nickel-NTA chromatography.

## Identification of PR-Ub sites

PR-Ub sites were identified by ETD-MS as previously described (Leidecker et al, 2016; Liu et al, 2021). GST-tagged STX17 and SNAP29 were modified by SdeA in vitro then denatured in 0.1 M Tris-HCl (pH 7.5) containing 8 M urea. The samples were washed three times in 200 μL of the same buffer in a 30-kDa Amicon Ultra 0.5-mL centrifugal filter (Merck) to remove free ubiquitin. The eluted protein was washed twice in 50 mM ammonium bicarbonate (pH 7.5) before digestion using a 1:50 ratio of trypsin the same buffer for 6 h. The peptides were desalted on a C18 column, followed by LC-MS/MS analysis. The spectra were collected and deconvoluted, and high-resolution ETD-MS/MS spectra were manually inspected to verify the sequence.

## Confocal microscopy and image analysis

Confocal images were observed using a Zeiss LSM780 microscope system fitted with a 63× 1.4 NA oil-immersion objective as well as argon and helium–neon ion lasers for the excitation of GFP (488 nm) and RFP (546 nm), respectively. For analyzing the RFP-GFP-LC3 colocalization with Legionella (stained with an Alexa647-tagged secondary antibody), images were analyzed in FIJI to determine the colocalization between 561 nm (RFP-LC3) and 633 nm (Legionella) channels. To count STX17, SNAP29 or LC3 puncta in cells, images were thresholded, converted to 8-bit and analyzed using the "Analyse particles" plugin in FIJI. Puncta of size 0.15–1.8 μm$^2$ (diameter 0.5–1.5 μm) were considered as autophagosomes (Mizushima et al, 2002). Area of LCVs was measured in FIJI from z-stack images of infected cells at 12 h.p.i. For checking recruitment of marker proteins to bacteria, the total number of intracellular bacteria per cell and the number of bacteria which is surrounded by the protein marker were counted manually in FIJI to find the % of intracellular bacteria which were positive for the recruitment of the protein. The data are means ± SEM of at least 50 cells taken from three independent experiments. For checking LAMP1 recruitment to intracellular bacteria (in Fig. 1E) only bacteria which are surrounded by LAMP1 staining were counted as positive for LAMP1. Partial colocalization of bacteria with LAMP1 puncta was often observed in ΔS Legionella infection. These were not counted as positive hits.

## Transmission electron microscopy (TEM)

At the indicated time points after infection (2 h or 6 h) with the respective Legionella strains at MOI:20, HeLa cells were washed two times with PBS buffer at RT then fixed using 2.5% glutaraldehyde in 0.1 M cacodylate buffer pH 7.2 for two hours at RT. Then, the cells were scraped from the Petri dish and collected by sedimentation. Post-fixation was done with 1% reduced osmium tetroxide at RT. Dehydration was performed by gradually increasing the acetone concentration. Samples were embedded using EPON. Ultrathin sections (50 nm) were cut using a Reichert Ultacut ultra-microtome and placed on a Formvar-coated slot grid. Grids were post-stained with uranyl acetate and lead citrate and analyzed using a digitalized Zeiss TEM 900 operated at 80 keV equipped with a Troendle 2 K camera.

## Immuno-electron microscopy

HeLa cells transfected with STX17-GFP and infected with Legionella-DsRed were fixed using freshly made 4% formaldehyde (FA) in 0.1 mol/L phosphate buffer (pH 7.4) by adding an equal amount of fixative to the medium for 5 min. Cells were post fixed using 2% FA in 0.1 mol/L phosphate buffer for 2 h and stored in 1% FA at 4 °C. Ultrathin cryosectioning and immunogold labeling were performed as previously described (Slot and Geuze, 2007). GFP-tagged syntaxin 17 was labeled using biotin-conjugated goat anti-GFP and rabbit anti-biotin antibodies from Rockland (610–4120, 600–106-215 and 100–4198, respectively). Antibodies were detected by protein A–10-nm gold particles (Cell Microscopy Core, Utrecht, the Netherlands).

## Legionella infection

Legionella pneumophila strains were obtained from Dr. Zhao-Qing Luo (Purdue University).

The strains used in the study were as follows:

WT-wild-type Legionella lp02; ΔS- Legionella lp02 with deletion of all 4 SidE effectors (ΔSdeA ΔSdeBΔSdeCΔSidE); ΔD- Legionella lp02 with deletion of both Dups (ΔDupAΔDupB); ΔR- Legionella lp02 with deletion of RavZ (ΔRavZ); ΔRΔS- Legionella lp02 with deletion of SidE and RavZ (ΔRavZΔSdeAΔSdeBΔSdeCΔSidE).

Legionella were grown for 3 days on N-(2-acetamido)-2-amino-ethanesulfonic acid (ACES)-buffered charcoal (BCYE) extract agar, at 37 °C, followed by growth for 20 h in CYE medium. Bacterial cultures (OD$_{600}$ = 3.2–3.6) were used to infect HEK 293 T, HeLa cells with a multiplicity of infection (MOI) of 10, and A549 cells with a MOI of 2. For 12-h infection experiments, we used a MOI of 1. HEK 293 T and HeLa cells were transfected with CD32 16 h before infection to facilitate bacterial entry.

## Intracellular replication of Legionella

A549 cells growing in 35-mm dishes were infected with Legionella strains at a MOI of 1. The infection was allowed to proceed for 90 min in antibiotic-free medium before switching to medium containing gentamycin for 60 min to kill the remaining extra-cellular bacteria. The cells were then lysed in 0.4% saponin immediately or after 24 or 48 h to release intracellular bacteria. Lysates were spotted onto BCYE plates at 1:10 and 1:100 dilutions. The intracellular bacterial load was assessed by counting bacterial colonies formed after 48 h. The number of colony-forming units (CFU) was calculated using the formula CFU/mL = (number of colonies × dilution factor)/volume of sample plated (mL). The fold-increase in CFU was calculated by normalizing the CFU at 24 or 48 h to that determined immediately after lysis.

## Proximity labeling of STX17

HeLa cells were transfected with APEX2-FLAG-STX17 and CD32 (for efficient infection with Legionella). 24 h after transfection, cells were

treated with 1ug/ml doxycycline and allowed to grow for 24 h to induce protein expression before infection with *Legionella* strains for different time periods. Before lysis, we added 500 μM of biotin-tyramide to the medium, incubated the cells for 1 h, and then treated them with 1 mM $H_2O_2$ for 1 min to trigger intracellular proximity-induced biotinylation. The reaction was quenched by washing the cells with PBS containing 10 mM sodium azide, 10 mM sodium ascorbate and 5 mM Trolox. The cells were lysed in 20 mM HEPES-KOH (pH 7.5) containing 150 mM KCl, 0.2 mM EDTA and 0.5% NP-40, then 1 mg of the lysate was incubated with 20 μL streptavidin-agarose resin overnight at 4 °C on a rotating platform. The pulled-down proteins were washed in lysis buffer and water three times and processed for western blotting and MS analysis. In the latter case, the beads were incubated in 8 M urea for 2 h at 37 °C before adding 10 mM DTT for reduction and 40 mM chloroacetamide for alkylation, followed by overnight digestion with 1 μg trypsin and 1 μg Lys-C. The peptides were then acidified with trifluoroacetic acid and desalted using Sep-Pak cartridges. Dried peptides were resuspended in TMT buffer and labeled by 6-plex TMT before MS analysis.

## Proximity labeling of bacterial vacuoles using ProtA-Turbo

HEK 293T cells were infected with *Legionella* strains for 2 h before proximity labeling as previously described (Santos-Barriopedro et al, 2023) with modifications. The cells were permeabilized with 0.05% digitonin in 20 mM HEPES/KOH (pH 7.5) containing 150 mM KCl and 2 mM $MgCl_2$ for 5 min at room temperature, then recovered by brief centrifugation (800×*g*, 5 min, room temperature) followed by two washes in the same buffer containing 0.02% digitonin. The cells were then incubated for 1 h at 4 °C in 500 μL of the same buffer without digitonin but containing 1 μg of the *Legionella*-specific antibody. After centrifugation and washing as above, the cells were incubated for 45 min at 4 °C with 1.5 μg ProtA-Turbo in 500 μL of the same buffer. Following another two washes as above, the cells were incubated with biotin reaction buffer (5 μM biotin, 5 mM $MgCl_2$, 1 mM ATP in 0.02% digitonin buffer) in a thermal shaker at 1000 rpm for 15 min at 37 °C. The cells were washed once with 0.5 mL of buffer without digitonin, lysed in 20 mM HEPES/KOH (pH 7.5) containing 150 mM KCl, 2 mM $MgCl_2$ and 1% Triton X-100, and then processed as described above for the streptavidin pulldown assay.

## Data availability

The mass spectrometry proteomics data have been deposited to the ProteomeXchange Consortium via the PRIDE partner repository with the Project accession. The data is available at https://www.ebi.ac.uk/pride/archive/projects/PXD058184. Other data are available upon request.

The source data of this paper are collected in the following database record: biostudies:S-SCDT-10_1038-S44318-025-00483-4.

## Peer review information

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

## Acknowledgements

RM received funding through an Alexander von Humboldt Stiftung postdoctoral fellowship. AB acknowledges funding from LYSOFOR2625. JK acknowledges funding from DFG-LYSOFOR2625 and the Dutch Research Council (NWO), project NEMI, number 184.034.014. ID acknowledges funding from the Deutsche Forschungsgemeinschaft (DFG; German Research Foundation) project number 259130777–SFB 1177, the European Research Council (ERC) under the European Union's Horizon 2020 research and innovation program (grant agreement number 742720), Else Kröner Fresenius Stiftung, Dr. Rolf M Schwiete Stiftung, and the Ernst Jung Prize for Medicine. IT has received funding from the European Union's Horizon 2020 research and innovation programme under the Marie Skłodowska-Curie grant agreement No 765445. The authors thank Sascha Martens for the mcherry-tagged PI3K (ATG14L) protein used in preliminary studies, the quantitative proteomics Unit of IBC2 for the use of their proteomics platform, and R Twyman, D Hoeller and A Gubas for critical reading of the manuscript.

## Author contributions

**Rukmini Mukherjee**: Conceptualization; Data curation; Formal analysis; Validation; Investigation; Methodology; Writing—original draft; Writing—review and editing. **Anshu Bhattacharya**: Conceptualization; Data curation; Formal analysis; Methodology. **Ines Tomaskovic**: Data curation; Formal analysis. **João Mello-Vieira**: Data curation. **Melinda Elaine Brunstein**: Resources; Methodology. **Marion Başoğlu**: Data curation; Formal analysis. **Tineke Veenendaal**: Resources; Data curation; Formal analysis. **Henry Bailey**: Data curation. **Thomas Colby**: Data curation. **Mohit Misra**: Resources; Data curation. **Stefan Eimer**: Resources; Data curation; Formal analysis; Supervision. **Judith Klumperman**: Resources; Data curation; Formal analysis; Writing—review and editing. **Christian Münch**: Resources; Supervision; Writing—review and editing. **Ivan Matic**: Resources; Data curation; Writing—review and editing. **Ivan Dikic**: Resources; Formal analysis; Supervision; Funding acquisition;

Investigation; Writing—original draft; Project administration; Writing—review and editing.

Source data underlying figure panels in this paper may have individual authorship assigned. Where available, figure panel/source data authorship is listed in the following database record: biostudies:S-SCDT-10_1038-S44318-025-00483-4.

## Funding

## Disclosure and competing interests statement
The authors declare no competing interests.

# Expanded View Figures

**Figure EV1. PR-Ub regulates autophagy in Legionella infection.** ▶

(**A**) SNARE proteins that were identified as PR-Ub substrates (Shin et al, 2020). (**B**) A549 cells were infected with different strains of *Legionella* for 2 h in the presence of 300 nM bafilomycin A1, then lysed and analyzed by western blot with antibodies against LC3 (and GAPDH as a loading control). The experiment was repeated three times with similar results. (**C**) HEK 293T cells were transfected with SdeA, its catalytic mutant SdeA (EE/AA) or a control vector for 16 h. Cells were treated with 100 nM Torin-1 as shown, then lysed and analyzed by western blot to check for LC3 levels. GAPDH was used as a loading control. Graph represents data from three independent experiments. Error bars indicate standard deviation. *P* value was calculated using two-tailed, type 3 Student's *t* test, **$0.01 \leq P < 0.001$. (**D**) HeLa cells were cotransfected with RFP-GFP-LC3 and HA-tagged SdeA/SdeA(EE/AA) or a control vector for 16 h before treatment with 300 nM Torin-1 for 4 h to induce autophagy. The cells were then fixed and incubated with an anti-HA antibody for confocal imaging. We counted the total number of puncta and the number of red puncta per cell in FIJI. The data are means ± SEM of 30 cells from three independent experiments (**$P = 0.002$ (vector vs SdeA), **$P = 0.041$(SdeA vs SdeA(EE/AA)). Scale bar: 5 µm. Dotted lines indicate cell outlines drawn from thresholding images in FIJI. (**E**) A549 cells were infected with the indicated strains of *Legionella*, and their intracellular replication was assessed at 0, 24 and 48 h post-infection. The data are means ± SEM of three independent experiments *$P = 0.012$ (WT vs ΔS), **$P = 0.00203$ (WT vs ΔRΔS). (**F**) RAW264.7 cells were infected with the indicated strains of *Legionella*, and their intracellular replication was assessed at 0, 24 and 48 h post-infection. The data are means ± SEM of three independent experiments. **$P = 0.0062$ (WT vs ΔS), **$P = 0.00072$ (WT vs ΔRΔS).

(A)

| Majority protein ID | Gene names | log2 enrichment factor dDup1 2 over dSidE | -Log10 p-value dDup1 2 over dSidE | log2 enrichment factor WT over dSidE | -Log10 p-value WT over dSidE |
|---|---|---|---|---|---|
| O00161 | SNAP23 | 4,29 | 2,41 | 3,94 | 2,21 |
| O60749 | SNX2 | 3,72 | 2,87 | 3,97 | 3,17 |
| P56962 | STX17 | 2,39 | 3,17 | 2,47 | 3,45 |
| Q9UNK0 | STX8 | 2,07 | 2,28 | 2,18 | 2,28 |
| O95721 | SNAP29 | 2,96 | 3,27 | 0,94 | 1,89 |
| Q9UNH7 | SNX6 | 1,83 | 2,08 | 2,17 | 2,08 |
| Q13190 | STX5 | 1,60 | 1,64 | 1,72 | 1,83 |

(B)

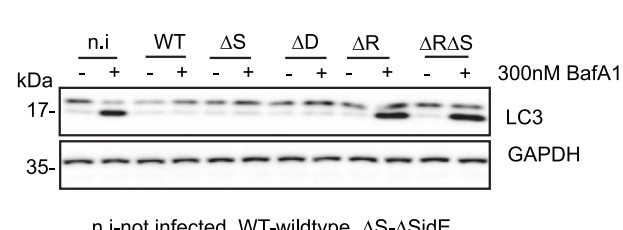

n.i-not infected, WT-wildtype, ΔS-ΔSidE,
ΔD-ΔDup, ΔR-ΔRavZ, ΔRΔS-ΔRavZΔSidE

(C)

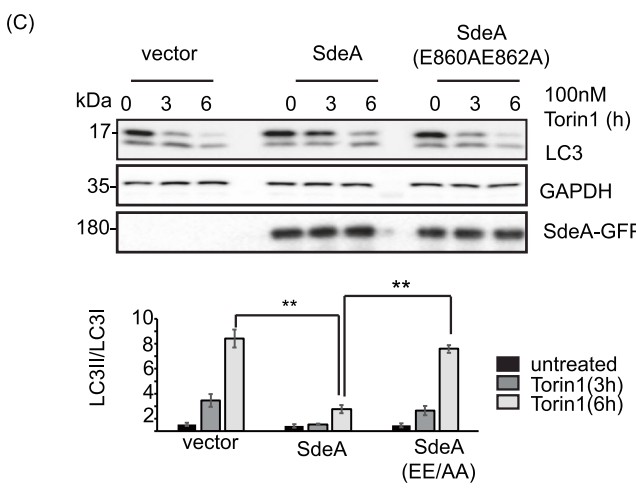

(D)

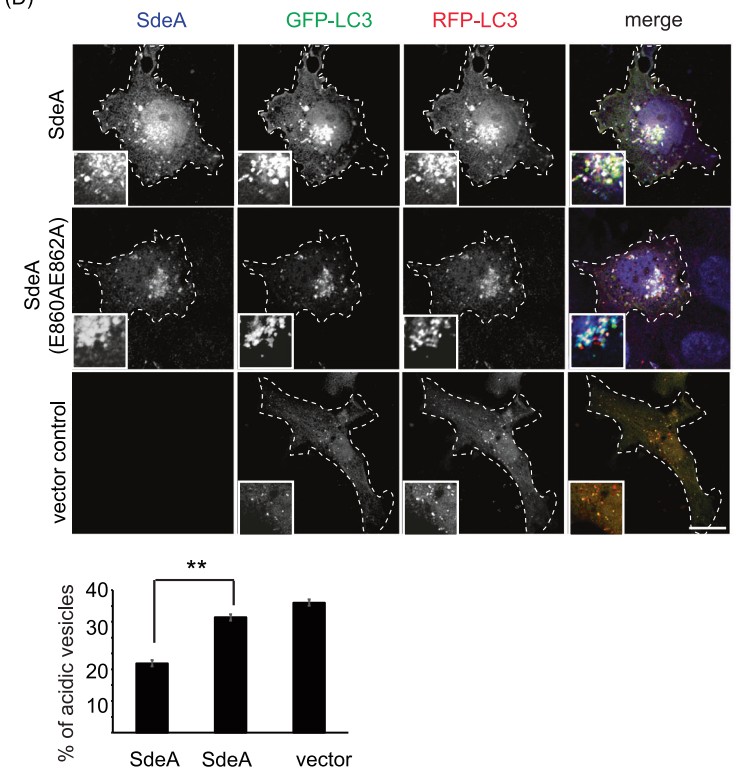

(E)

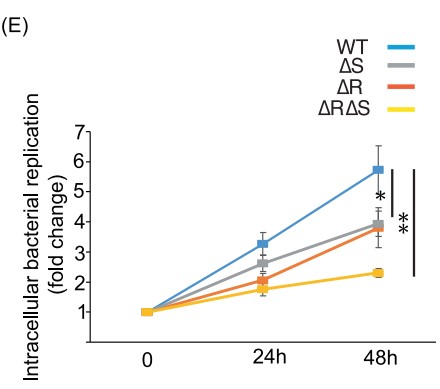

(F)

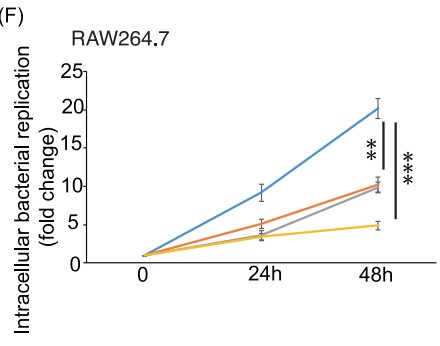

(A)

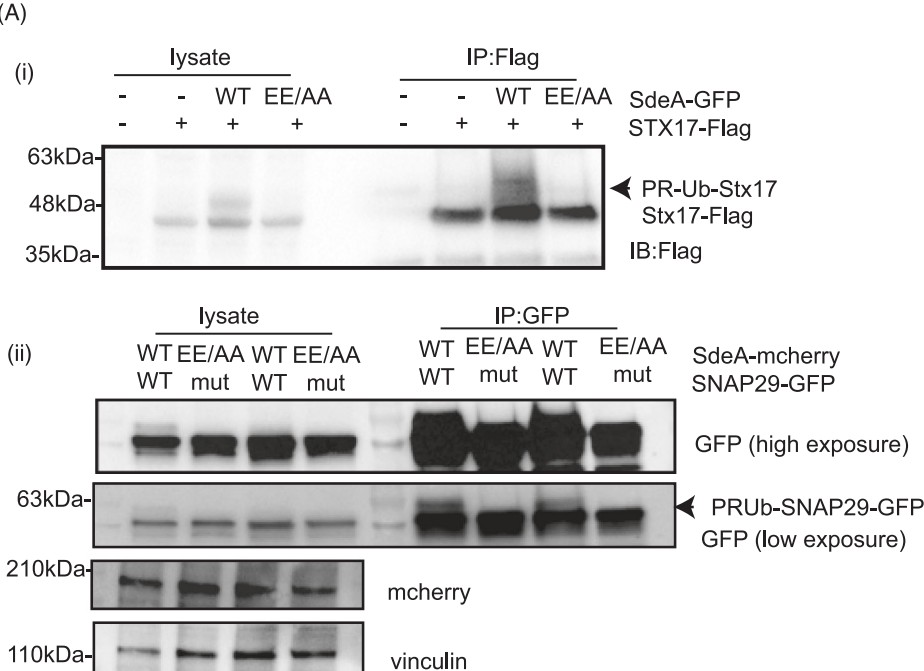

(B)

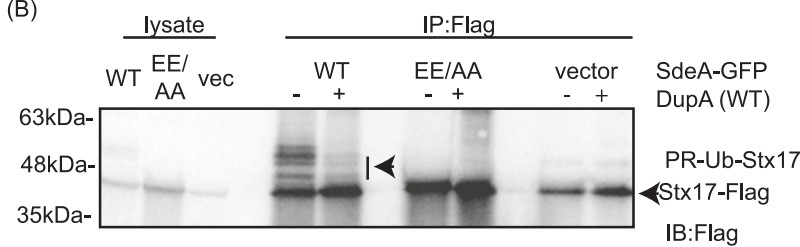

(C)

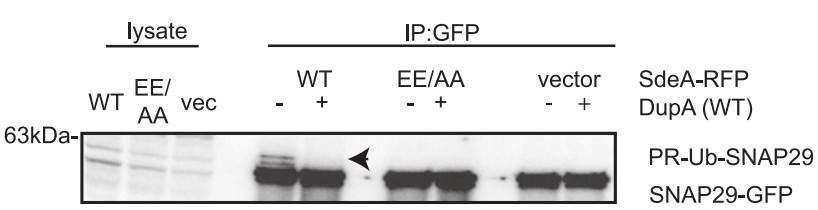

(D)                                    (E)

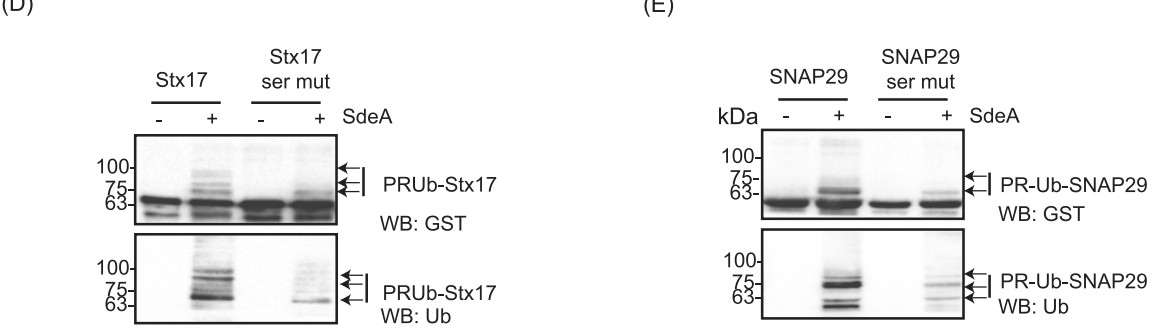

◀ **Figure EV2. STX17 and SNAP29 are modified by PR-Ub on specific serine residues during Legionella infection.**

(A) (i) HEK 293 T cells were cotransfected with FLAG-STX17 and GFP-tagged SdeA/SdeA(EE/AA) or a control vector for 16 h. FLAG-STX17 was immunoprecipitated using FLAG resin and analyzed by western blot using antibodies against FLAG to detect PR-Ub-modified and unmodified FLAG-STX17. The experiment was repeated 2 times with similar results. (ii) HEK 293T cells were cotransfected with FLAG-STX17 and GFP-tagged SdeA/SdeA(EE/AA) or a control vector and immunoprecipitated as shown in (B). The samples were then treated with or without pure DupA for 1 h before western blotting with antibodies against FLAG to detect PR-Ub-modified and unmodified FLAG-STX17. The experiment was repeated two times with similar results. (B) HEK 293T cells were cotransfected with GFP-SNAP29 and HA-tagged SdeA/SdeA(EE/AA) or a control vector for 16 h. GFP-SNAP29 was immunoprecipitated with anti-GFP beads, treated with or without pure DupA for 1 h and analyzed by western blot with antibodies against GFP to detect PR-Ub-modified and unmodified SNAP29. The experiment was repeated two times with similar results. (C) GST-STX17 and GST-STX17(S195AS202AS209A) were incubated with or without SdeA in the presence of 1 mM NAD$^+$ and ubiquitin for 1 h. The samples were analyzed by western blot using antibodies against ubiquitin and GST to detect PR-Ub. The experiment was repeated three times with similar results. (D) GST-STX17 and its PR-Ub-deficient mutant (S195AS202AS209A) were modified with or without SdeA, in the presence of 1 mM NAD+ and ubiquitin for 1 h. Samples were analyzed by western blot with antibodies against ubiquitin and GST to detect PR-Ub. The experiment was repeated three times with similar results. SdeA(EE/AA): mART mutant SdeA(E860AS862A) (E) GST-SNAP29 and its PR-Ub-deficient mutant (S61AS63AS70A) were modified with or without SdeA, in the presence of 1 mM NAD$^+$ and ubiquitin for 1 h. Samples were analyzed by western blot with antibodies against ubiquitin and GST to detect PR-Ub. The experiment was repeated three times with similar results. SdeA(EE/AA): mART mutant SdeA(E860AS862A).

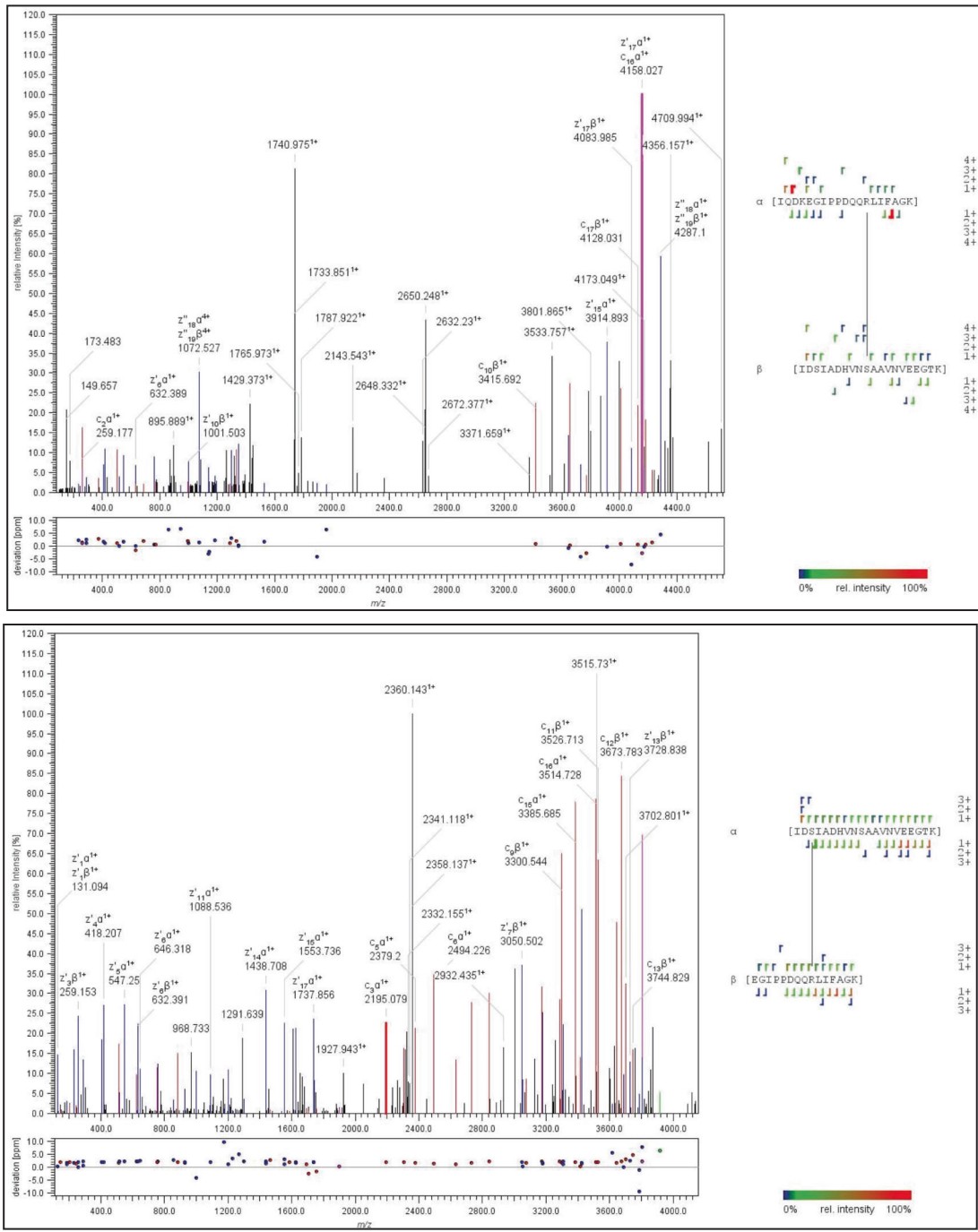

MS spectra of PR-Ub modified STX17(1-224)

**Figure EV3. Identification of STX17 residues modified by PR-Ub.**

Mass spectra of PR-Ub-modified STX17(1–224). S202, S209 were identified as the modified serine residues by high-resolution ETD mass spectrometry.

(A)

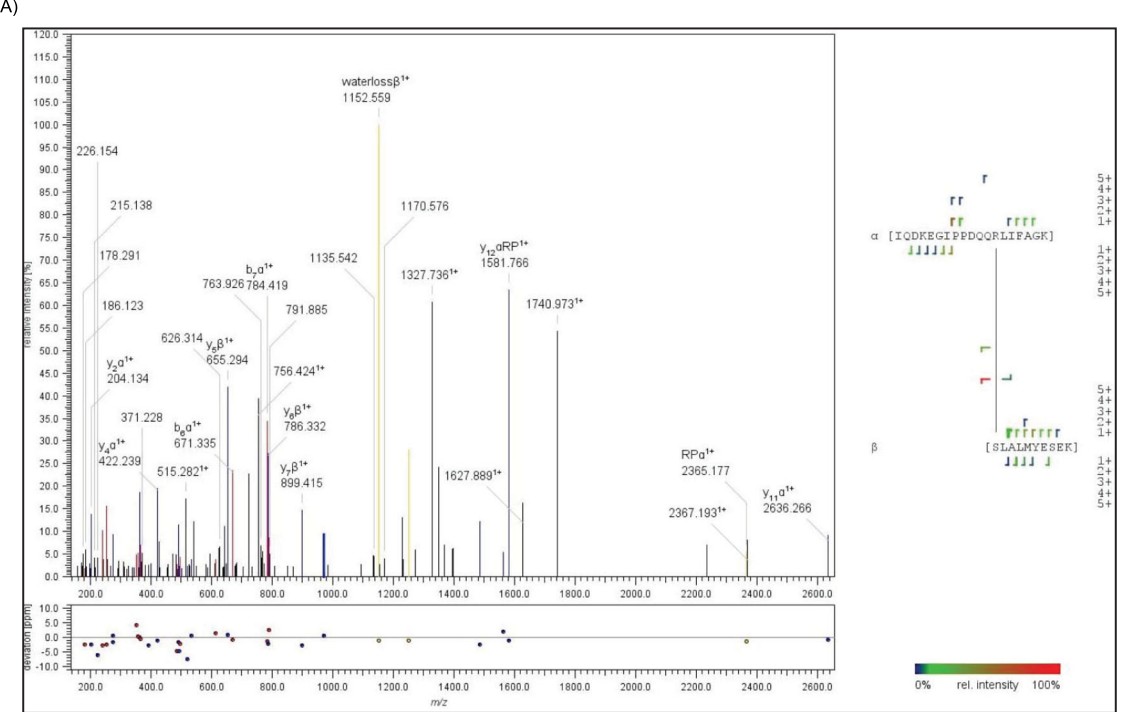

MS spectra of PR-Ub modified SNAP29

(B)

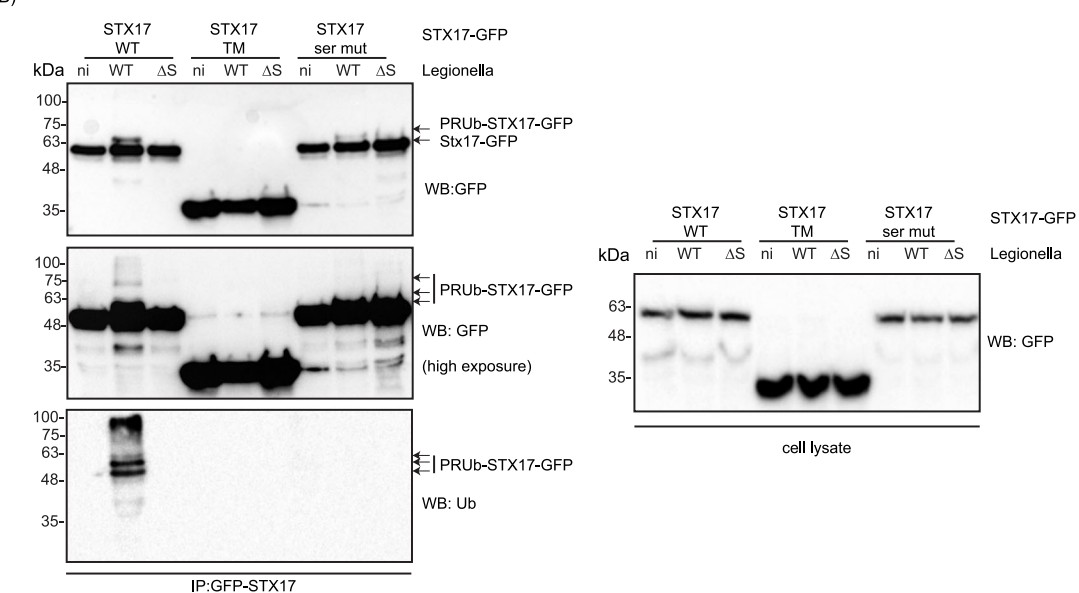

**Figure EV4. Identification and validation of serine residues on STX17 and SNAP29 that are modified by PR-Ub.**

(A) Mass spectrum and deduced sequence map of PR-Ub-modified SNAP29. (B) HEK 293 T cells were transfected with GFP-tagged WT STX17, STX17TM, or the STX17 serine mutant (S195AS202AS209A) followed by *Legionella* infection for 2 h. STX17 was then immunoprecipitated using anti-GFP beads followed by western blotting with antibodies against GFP and ubiquitin. Cell lysates were analyzed by western blot with an antibody against GFP to check the expression levels of GFP-STX17 constructs. The experiment was repeated three times with similar results.

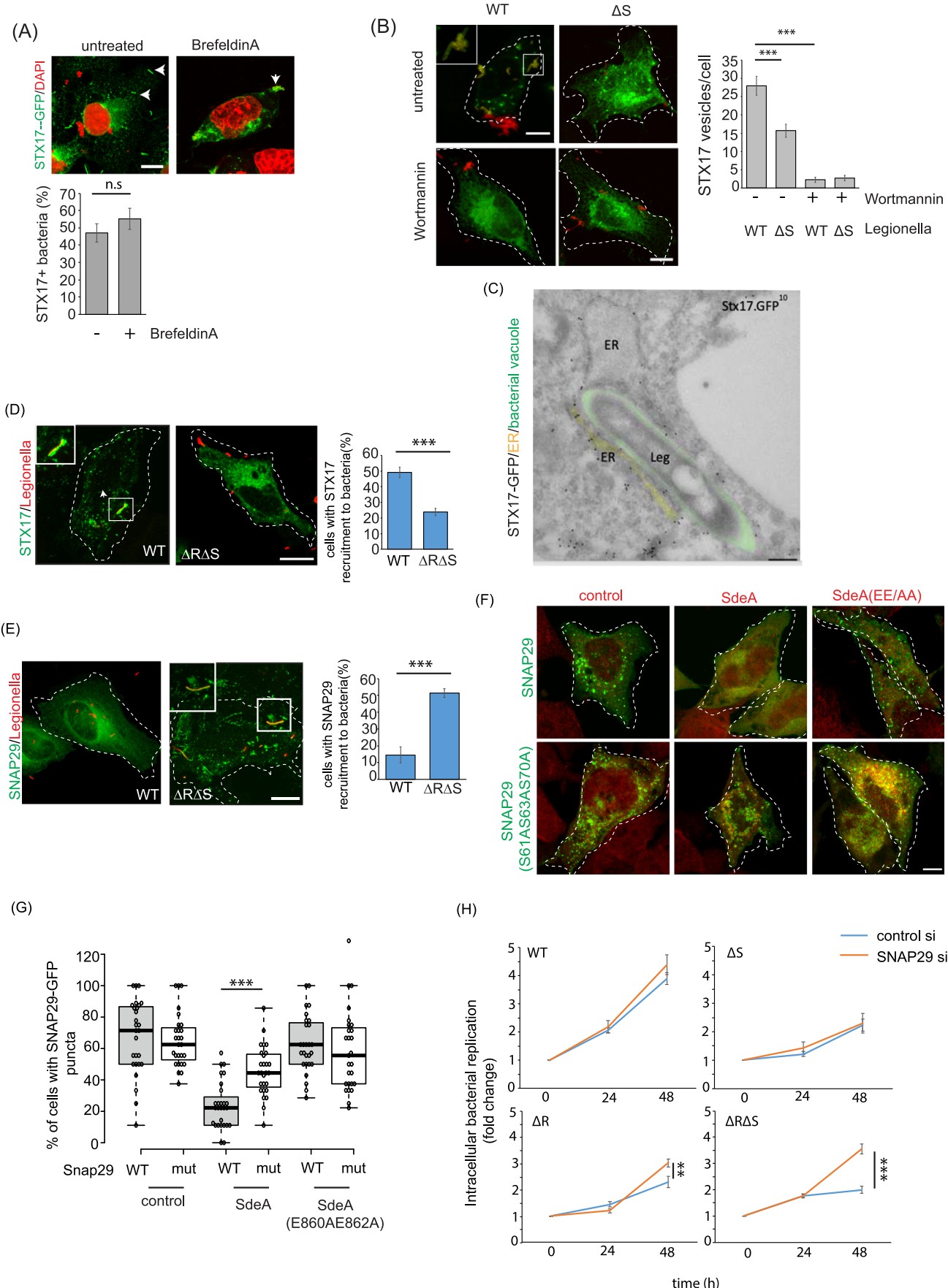

◀ **Figure EV5. Recruitment of STX17 and SNAP29 to bacterial vacuoles is regulated by their PR-Ub.**

(A) A549 cells expressing STX17-GFP were infected with WT *Legionella* for 2 h in the presence or absence of 100 nM brefeldin A before fixation and staining the intracellular bacteria by DAPI. STX17-positive bacteria was counted in 50 cells per set, taken from three independent experiments. Error bars indicate SEM. Difference between sets was non-significant from p value calculated by two-tailed, type 3 Student's *t* test. Scale bar:10 μm. (B) A549 cells expressing STX17-GFP were infected with *Legionella* (WT/ΔS) for 2 h in the presence or absence of 100 nM wortmannin before fixation and immunostaining with antibodies against *Legionella*. p value was calculated by two-tailed, type 3 Student's *t* test. \*\*\**P* = 4.45E-6(WT and ΔS sets without wortmannin), \*\*\**P* = 2.05E-5 (WT +/-wortmannin), Graph represents *n* = 50 cells taken from three experiments, error bars indicate SEM. Scale bar: 5 μm. Dotted lines indicate cell outlines drawn from thresholding images in FIJI. (C) Immuno-electron microscopy of HeLa cells transfected with STX17-GFP and infected with WT Legionella-DsRed for 4 h. Ultrathin cryosection immunogold labeled for STX17-GFP by protein A–10-nm gold. Colors are added by Photoshop: Yellow marks a STX-17.GFP-positive ER cisterna closely aligned with the Legionella (Leg) containing vacuole. Green marks the space between the vacuolar membrane and enclosed Legionella. Bar, 200 nm. (D) A549 cells were infected with WT or ΔRΔS *Legionella* for 1 h, fixed and immunostained with the STX17 and Legionella antibodies to check for the recruitment of STX17 to intracellular bacteria. White arrows mark intracellular bacteria with STX17 recruitment. The data are means ± SEM of 118 cells from three independent experiments. p value was calculated by two-tailed, type 3 Student's *t* test. \*\*\**P* = 4.21E-6. Scale bar: 5 μm. Dotted lines indicate cell outlines drawn from thresholding images in FIJI. (E) A549 cells were infected with WT or ΔRΔS *Legionella* for 1 h, fixed and immunostained with the SNAP29 and Legionella antibodies to check for the recruitment of SNAP29 to intracellular bacteria. White arrows mark intracellular bacteria with SNAP29 recruitment. The data are means ± SEM of 120 cells from three independent experiments. p value was calculated by two-tailed, type 3 Student's *t* test. \*\*\**P* = 2.21E-4. Scale bar: 5 μm. Dotted lines indicate cell outlines drawn from thresholding images in FIJI. (F) HeLa cells were cotransfected with RFP-tagged SdeA or its catalytic mutant (E860AE862A) and GFP-tagged WT SNAP29 or its PR-Ub-deficient mutant. Cells were treated with 300 nM Torin-1 for 4 h to induce autophagy before fixation and confocal imaging. Scale bar:5 μm. Dotted lines indicate cell outlines drawn from thresholding images in FIJI. (G) The graph shows the number of cells with SNAP29-GFP puncta (from panel d) counted in FIJI. In the box plot, center lines show the medians; box limits indicate the 25th and 75th percentiles as determined by R software; whiskers extend 1.5 times the interquartile range from the 25th and 75th percentiles. n > 30 cells taken from three independent experiments. *P* value was calculated using two-tailed, type 3 Student's *t* test, \*\*\**P* = 0.00032. In bar graph, the data are means ± SEM of n > 30 cells from three independent experiments. Scale bar:5 μm. (H) A549 cells were treated with SNAP29 or control siRNA for 48 h followed by infection Legionella. Intracellular bacterial replication was assessed after 0, 24 and 48 h. Data are means ± SEM of three independent experiments. *P* value was calculated using two-tailed, type 3 Student's *t* test, \*\*\**P* = 2.1E-4 (ΔR), \*\**P* = 0,.031(ΔRΔS). (ni not infected, WT wild-type, *Legionella*, ΔS-ΔSidE *Legionella*).

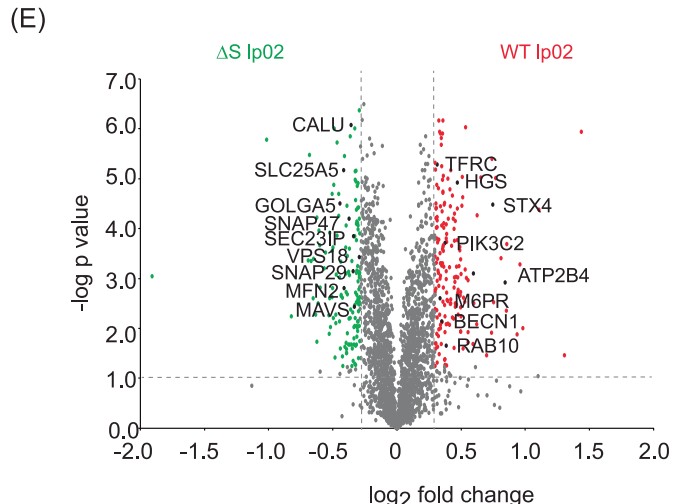

(A)

2 h p.i    6 h p.i

WT Legionella infection

Inset

Lyl

(B)

FLAG/DAPI

APEX2-FLAG-STX17 in
WT Legionella infection

(C)

ni  WT  ΔS        ni  WT  ΔS        ni  WT  ΔS

FIP200            WIPI2             LC3B

ULK1              ATG5              SNAP29

ATG13             ATG12             STX17

ATG14L            ATG16L            VAMP8

Beclin1

(D)

uninfected                    WT lp02

VDAC3
RPL35A
RPL9                          RAB1A
                              TFRC
FUS                           ATG16L1
COPB2                         RAB5C
SNAP29                        ATPV1A
                              ATPV0C
CLINT1                        ARF3 SAR1
SRP9  SIGMAR1                 BECN1
                              SEC23A
                              PIK3C2G
                              ATG3

-log p value vs log₂ fold change

Intracellular pathways enriched
upon WT Legionella infection

Secretory pathway

ARF3, ARF4, RAB1A, RAB1B,RAB6A,
RAB9A, RAB34, COPG1, COPG2,
SEC23A,STT3A, STT3B, B4GALT1,
LMAN2, SAR1A, YIF1A, TMED5, ERGIC1

Autophagy

ATG3, ATG16L, BECN1,PARK7,
PERILIPIN

Endocytic pathway

HGS1, RAB8A, RAB5A, RAB5C,
RAB11A, RAB35, GAPVD1, VPS4A,
SNX6, SCARB2, TFRC

Proteins enriched in WT

Secretory pathway

ARF6, ARF1, RAB10, RAB11,RAB1A,
M6PR, AP1M1, AP3S1

Autophagy

PIK3C3, PIK3C2, ATG16L, BECN1,PARK7

Endocytic pathway

TFRC, HGS, AP2A2, AP2M1,AP2B1, SNX6

Docking at plasma membrane

ATP2B4, STX4, STXBP3, CD81, CD99,
ATP1B1,RAB3D

(E)

ΔS lp02                       WT lp02

CALU
SLC25A5                       TFRC
                              HGS
GOLGA5                        STX4
SNAP47
SEC23IP                       PIK3C2
VPS18
SNAP29                        ATP2B4
MFN2                          M6PR
MAVS                          BECN1
                              RAB10

-log p value vs log₂ fold change

Proteins enriched in ΔS

Golgi and ERGIC

GOLGA5, GOLGA2, GOLGB1, GOLGA8F,
GOPC, CALU, LMAN1, LMAN2,SEC23B,
SEC24A, SEC24C,SEC31A, ERGIC2, BET1,
GLG1,STX5

SNARES and  fusion with lysosome

SNAP29, SNAP47, VPS41, VPS11,VPS18

Mitochondria & MAMs

MICOS, TIMM21, UQCRH, MFN2, MAVS

◀  **Figure EV6.  Proximity labeling of STX17 during *Legionella* infection.**

(A) HeLa cells expressing CD32 (for efficient uptake of *Legionella*) were infected with WT Legionella for 2 h or 6 h. Cells were fixed using 2.5% glutaraldehyde in 0.1 M cacodylate buffer for two hours at RT. Cells were scraped of the petridish, post-fixed with 1% reduced osmium tetroxide at RT, dehydrated and embedded using EPON. Ultrathin sections (50 nm) were imaged by transmission electron microscopy. Arrows mark intracellular bacteria, Lyl-lysosome-like organelles. (B) HeLa cells expressing APEX2-FLAG-STX17 are infected with WT *Legionella* for 2 h followed by fixing cells and staining cells with FLAG antibody to check its recruitment to intracellular bacteria. DAPI marks nuclear DNA and cytosolic bacteria. Scale bar:5 μm. White arrows indicate intracellular bacteria. (C) Lysates used as input in streptavidin IP shown in Fig. 4G. (D) Volcano plot showing how the biotin-labeled proteome changes when HeLa cells expressing APEX-STX17 are infected with WT *Legionella* for 2 h; GO analysis of the biotin-labeled proteome showing pathways upregulated by infection with WT *Legionella*. Red and green indicate compartments containing proteins enriched following infection with WT *Legionella* and in uninfected cells, respectively. Data represent mean fold change of three experimental replicates per infection set ($n = 3$). P value was calculated using two-tailed type 3 Student's *t* test and significant candidates were chosen having P value ≤ 0.01 and log2(fold change) value minimum of ±0.5. (E) Volcano plot showing changes in the biotin-labeled proteome following the infection of HeLa cells expressing APEX-STX17 with WT and ΔS *Legionella* for 2 h. GO analysis of the biotin-labeled proteome showing pathways upregulated by infection with WT vs ΔS *Legionella*. Data represents mean fold change of three experimental replicates per infection set ($n = 3$). P value was calculated using two-tailed type 3 Student's *t* test and significant candidates were chosen having P value ≤ 0.01 and log2 (fold change) value minimum of ±0.5. Red and green indicate compartments containing proteins enriched following infection with WT and ΔS *Legionella*, respectively. (ni not infected, WT wild-type *Legionella*, ΔS-ΔSidE).

