## [Peer Review File · The EMBO Journal]

Phosphoribosyl ubiquitination of SNARE proteins regulates autophagy during Legionella infection

Rukmini Mukherjee, Anshu Bhattacharya, Ines Tomaskovic, João Mello-Vieira, Melinda Brunstein, Marion Basoglu, Tineke Veenendaal, Henry Bailey, Thomas Colby, Mohit Misra, Stefan Eimer, Judith Klumperman, Christian Münch, Ivan Matic, and Ivan Dikic

Corresponding author(s): Ivan Dikic (dikic@biochem2.uni-frankfurt.de)

Review Timeline:

Submission Date:	25th Feb 24
Editorial Decision:	27th Mar 24
Revision Received:	9th Jan 25
Editorial Decision:	11th Mar 25
Revision Received:	25th Apr 25
Accepted:	15th May 25

Editor: William Teale

Transaction Report:

Dear Ivan,

Thank you again for the submission of your manuscript entitled "Serine ubiquitination of SNAREs generates bacterial vacuoles and prevents their lysosomal fusion" and for your patience during the review process. Your manuscript was sent to two referees; we have now received the reports from both of them, which I copy below.

As you can see from their comments, while the referees need a more careful explanation of the rationale behind aspects of your experimental design and more careful controls in some of the experiments you present, both point out the potential value of your work to the scientific community.

Based on the overall interest expressed in the reports, therefore, I would like to invite you to address the comments of all referees in a revised version of the manuscript. I should add that it is The EMBO Journal policy to allow only a single major round of revision and that it is therefore important to resolve the main concerns at this stage. I believe the concerns of the referees are reasonable and addressable, but please contact me if you have any questions, need further input on the referee comments or if you anticipate any problems in addressing any of their points. I am happy to arrange a Zoom call after Easter once you have had a chance to digest the reports. Please, follow the instructions below when preparing your manuscript for resubmission.

I would also like to point out that as a matter of policy, competing manuscripts published during this period will not be taken into consideration in our assessment of the novelty presented by your study ("scooping" protection). We have extended this 'scooping protection policy' beyond the usual 3 month revision timeline to cover the period required for a full revision to address the essential experimental issues. Please contact me if you see a paper with related content published elsewhere to discuss the appropriate course of action.

Again, please contact me at any time during revision if you need any help or have further questions.

Thank you very much again for the opportunity to consider your work for publication. I look forward to your revision.

Best regards,

William

William Teale, Ph.D.
Editor
The EMBO Journal

When submitting your revised manuscript, please carefully review the instructions below and include the following items:

- 1) a .docx formatted version of the manuscript text (including legends for main figures, EV figures and tables). Please make sure that the changes are highlighted to be clearly visible.
- 2) individual production quality figure files as .eps, .tif, .jpg (one file per figure).
- 3) a .docx formatted letter INCLUDING the reviewers' reports and your detailed point-by-point response to their comments. As part of the EMBO Press transparent editorial process, the point-by-point response is part of the Review Process File (RPF), which will be published alongside your paper.
- 4) a complete author checklist, which you can download from our author guidelines ([https://wol-prod-cdn.literatumonline.com/pb-assets/embo-site/Author Checklist%20-%20EMBO%20J-1561436015657.xlsx](https://wol-prod-cdn.literatumonline.com/pb-assets/embo-site/Author%20Checklist%20-%20EMBO%20J-1561436015657.xlsx)). Please insert information in the checklist that is also reflected in the manuscript. The completed author checklist will also be part of the RPF.
- 5) Please note that all corresponding authors are required to supply an ORCID ID for their name upon submission of a revised manuscript.
- 6) We require a 'Data Availability' section after the Materials and Methods. Before submitting your revision, primary datasets produced in this study need to be deposited in an appropriate public database, and the accession numbers and database listed

under 'Data Availability'. Please remember to provide a reviewer password if the datasets are not yet public (see <https://www.embopress.org/page/journal/14602075/authorguide#datadeposition>). If no data deposition in external databases is needed for this paper, please then state in this section: This study includes no data deposited in external repositories. Note that the Data Availability Section is restricted to new primary data that are part of this study.

Note - All links should resolve to a page where the data can be accessed.

8) For data quantification: please specify the name of the statistical test used to generate error bars and P values, the number (n) of independent experiments (specify technical or biological replicates) underlying each data point and the test used to calculate p-values in each figure legend. The figure legends should contain a basic description of n, P and the test applied. Graphs must include a description of the bars and the error bars (s.d., s.e.m.).

9) We would also encourage you to include the source data for figure panels that show essential data. Numerical data can be provided as individual .xls or .csv files (including a tab describing the data). For 'blots' or microscopy, uncropped images should be submitted (using a zip archive or a single pdf per main figure if multiple images need to be supplied for one panel). Additional information on source data and instruction on how to label the files are available at .

10) We replaced Supplementary Information with Expanded View (EV) Figures and Tables that are collapsible/expandable online (see examples in <https://www.embopress.org/doi/10.15252/embj.201695874>). A maximum of 5 EV Figures can be typeset. EV Figures should be cited as 'Figure EV1, Figure EV2" etc. in the text and their respective legends should be included in the main text after the legends of regular figures.

12) Our journal encourages inclusion of *data citations in the reference list* to directly cite datasets that were re-used and obtained from public databases. Data citations in the article text are distinct from normal bibliographical citations and should directly link to the database records from which the data can be accessed. In the main text, data citations are formatted as follows: "Data ref: Smith et al, 2001" or "Data ref: NCBI Sequence Read Archive PRJNA342805, 2017". In the Reference list, data citations must be labeled with "[DATASET]". A data reference must provide the database name, accession number/identifiers and a resolvable link to the landing page from which the data can be accessed at the end of the reference. Further instructions are available at .

Further instructions for preparing your revised manuscript:

At EMBO Press we ask authors to provide source data for the main manuscript figures. Our source data coordinator will contact you to discuss which figure panels we would need source data for and will also provide you with helpful tips on how to upload

and organize the files.

We realize that it is difficult to revise to a specific deadline. In the interest of protecting the conceptual advance provided by the work, we recommend a revision within 3 months (25th Jun 2024). Please discuss the revision progress ahead of this time with the editor if you require more time to complete the revisions. Use the link below to submit your revision:

Referee #1:

Mukherjee and Bhattacharya et al. show that phosphoribosyl-linked ubiquitylation of substrates by Legionella SidE-family proteins blocks autophagic flux and protects Legionella from degradation by the host. Mukherjee and Bhattacharya utilize a number of techniques to show that SNARE proteins STX17 and SNAP29 are modified by SidE family members upon infection with Legionella. They show that modification of STX17 leads to its recruitment to nascent membranes which form the LCVs, where they suggest it recruits other LCV-associated proteins. Meanwhile they present data showing modification of SNAP29 with PR-Ub blocks formation of the SNARE complex required for fusion of STX17-positive vesicles with the lysosome. Whilst they demonstrate a pathway where Legionella avoid xenophagy, the manuscript lacks mechanistic insights to fully explain the data and therefore limits the interest forecast for this work.

Major Concerns:

- 1) Other examples of PR-ubiquitylation result in a steric block in protein function, similar to the model presented for SNAP29. How can the authors explain the enhanced function exhibited by PR-ubiquitylated STX17? This mechanism would be an important addition to the story.
- 2) It is unclear why some experiments use Torin1 and others do not. The authors should make this more consistent or otherwise explain their rationale.
- 3) Does the growth defect observed in the ΔS strain arise specifically from the role in blocking autophagy? Figure 4C-D speak to this somewhat, but to what extent does SNAP29 contribute to this effect?
- 4) In the Legionella replication assays using A549 cells, are the bacteria really only dividing once or twice within the 2-day time period? Perhaps a more relevant macrophage model would provide a better measure.
- 5) The authors include some analysis of DupA and DupB in Figure 1, but do not build that into the model for SidE regulation of autophagy. Could they be downregulating the effects of the SidE ligases? For example, in the experiment shown in Figure 1C, in the background of the ΔR mutant, would DupA/B deletion or overexpression alter the number of acidified vesicles or LC3-positive LCVs?
- 6) To what extent are the SidE family members acting redundantly in this process? Some experiments refer only to SidE, while others use SdeA. Is the ΔS Legionella strain lacking all four effectors? This is unclear from the text.
- 7) The methods section requires additional details. Specifically, more information should be provided on cell culture methods, the construction of the APEX cell line, the details of all Legionella strains used in the study, quantitation of microscopy images (especially for LAMP1 staining, as well as details on the number of LCVs quantified), and the analysis of mass spectrometry data.
- 8) The proteomics datasets should be deposited in an appropriate database and made publicly available.
- 9) Some of the representative microscopy images shown are counter to what would be expected. For example, in the WT infection shown in Figure 1B, where RavZ should be present, why are there so many LC3-positive LCVs? Also, in Figure 1E, for the ΔS mutant, there appears to be colocalization of LAMP1 and Legionella, even though the argument made in the text is that

there is none.

- 10) More analysis should be performed on the $\Delta R\Delta S$ double mutant, including quantitation of LC3-positive LCVs to show that SidE is acting downstream of this step, as well as additional EM to show fusion of these double mutant LCVs with the lysosome.
- 11) The type of ubiquitin linkage that SidE ligases catalyze is not explained clearly, and categorizing it as serine ubiquitylation is very confusing for the reader. Improved clarity in the text, as well as the schematics in Figure 1A, would be very helpful. The authors should also consider distinguishing this from conventional serine ubiquitylation with an alternate nomenclature, possibly PR-ubiquitylation.
- 12) Can the authors justify their choice of utilizing A549 cells as a model for Legionella infection?
- 13) Some panels, such as Figure 2G, are missing statistical analyses.
- 14) Input samples should be shown for the streptavidin IP experiments presented in Figure 4G. Some sort of control showing that the STX17-APEX construct localizes to the LCV should also be included.
- 15) There is no clear indication of how many times certain experiments were repeated, particularly for the western blots that also lack quantitation.

Minor Concerns:

- 1) The naming of Legionella strains is not consistent throughout. Some figures use "WT", while others use "Lp02".
- 2) The magnified inset for Figure 2F is not indicated properly.
- 3) Some western blots are lacking labels, such as the top portion of Figure 2A. The experiments presented in Figures 5C and 5F are very difficult to interpret as currently labeled (some bands are not labeled, some columns are listed as "P" and others as "PR", there is no indication of what SNAP29 48- means).
- 4) The presentation of the EM images could be improved. The empty space could be used to show a representative lysosome or multivesicular body. The labels on top of the image, such as "LkL", are very difficult to read and should be replaced with arrows.
- 5) Different time points of infection are used for EM across WT and ΔS infections. The authors should make this consistent or otherwise rationalize their decisions.
- 6) The experimental schematic in Figure 4E should be altered to include the other sample comparisons studied.

Referee #2:

Mukherjee and co-workers report that Legionella SidE effectors mediate phosphor-ribosyl-linked serine ubiquitination (PR-Ub) of the autophagic SNARE proteins STX17 and SNAP29. They find that PR-Ub of STX17 increases its affinity for the ER-binding PI3K subunit ATG14, thereby promoting recruitment of ER membranes to the bacterial vacuole in a PI3K-dependent manner and the formation of bacterial replicative vacuoles. The authors also find that PR-Ub of SNAP29 prevents formation of the STX17-SNAP29-VAMP8 SNARE complex that drives fusion of autophagosomes with lysosomes, thereby preventing fusion of STX17-containing bacterial vacuoles with lysosomes. This manuscript is conceptually interesting since it indicates a novel mechanism for how a bacterium produces factors that sustain its replicative niche. Overall the experimental data are of high quality and support the proposed mechanisms. I therefore think this manuscript would be a good fit for EMBO Journal pending successful revision.

Major point:

In a recent paper, Jian et al. reported that a SNARE complex that contains SNAP47 instead of SNAP29 mediates autophagosome-lysosome fusion in both bulk and selective autophagy (Cell Research, 2024). From Figure 4F it appears that SNAP47 is not modified by PR-Ub, and it remains unclear why SNAP47 cannot substitute for SNAP29 in vacuole-lysosome fusion in Legionella infected cells when the latter is inhibited by PR-Ub. The authors should investigate this issue.

Mukherjee and Bhattacharya et al. show that phosphoribosyl-linked ubiquitylation of substrates by Legionella SidE-family proteins blocks autophagic flux and protects Legionella from degradation by the host. Mukherjee and Bhattacharya utilize a number of techniques to show that SNARE proteins STX17 and SNAP29 are modified by SidE family members upon infection with Legionella. They show that modification of STX17 leads to its recruitment to nascent membranes which form the LCVs, where they suggest it recruits other LCV-associated proteins. Meanwhile they present data showing modification of SNAP29 with PR-Ub blocks formation of the SNARE complex required for fusion of STX17-positive vesicles with the lysosome. Whilst they demonstrate a pathway where Legionella avoid xenophagy, the manuscript lacks mechanistic insights to fully explain the data and therefore limits the interest forecast for this work.

Major Concerns:

1) Other examples of PR-ubiquitylation result in a steric block in protein function, similar to the model presented for SNAP29. How can the authors explain the enhanced function exhibited by PR-ubiquitylated STX17? This mechanism would be an important addition to the story.

This is an important point raised by the reviewer. From the data presented in this manuscript and in other studies studying autophagy in pathogen infection, it is evident that pathogenic infection activates the autophagy response of the host which tries to counteract the infection through lysosomal degradation. During Legionella infection the bacterial RavZ delipidates ATG8 proteins to prevent formation of conventional ATG8+ autophagosomes (Choy et al., 2012). However, we observed that pre-autophagosome like compartments are formed from the ER in a STX17 dependent manner. The site of PR-ubiquitylation on STX17 is S202 which coincides with the phosphorylation site of STX17 by TBK1. Phosphorylation of STX17 at S202 increases formation of FIP200-ATG13 pre-autophagosomal structures (Kumar et al., 2019). These early autophagy markers are found on the STX17 positive bacteria making it plausible to hypothesize that in this case PR-ubiquitylation of S202 has a similar effect as its phosphorylation.

On a different note, studies of ubiquitylation of other ER proteins, for example the FAM134 family of proteins have showed how ubiquitylation can influence membrane curvature, and facilitate clustering of receptors which is instrumental in remodeling membranes during ER-phagy (Gonzalez et al., 2023). Interestingly, like STX17 we see that PR-Ubiquitinated FAM134B and FAM134C are also recruited to bacterial vacuoles during Legionella infection (Shin et al., 2020). Though Syntaxins are structurally very different from the reticulon homology domain (RHD) containing FAM proteins we are in the process of performing a study to understand how PR-ubiquitylation and ubiquitination of Syntaxins may affect its function during Legionella infection and in physiology/pathophysiology.

The STX17+ATG8- preautophagosome like bacterial vacuoles that we observe are similar to STX17 positive autophagosomes in ATG3KO cells which lack the means of lipidating their autophagosomes. These vesicles are more elliptical than conventional ATG8 positive autophagosomes, have slower fusion dynamics with lysosomes and

reduced degradation of the inner autophagosomal membrane after autophagosome-lysosome fusion (Tsuboyama et al., 2016).

2) It is unclear why some experiments use Torin1 and others do not. The authors should make this more consistent or otherwise explain their rationale.

Torin1 was used in experiments to induce conventional autophagy where the role of LC3 lipidation (formation of LC3+ autophagosomes) was measured: - In figure 1 (panels B, C, and D) and figure 2 (panels D and E). For all other experiments, studying the role of STX17 and SNAP29 PR-ubiquitination in bacterial infection [where the effector RavZ blocked LC3B lipidation], Torin1 was not used to induce autophagy.

3) Does the growth defect observed in the ΔS strain arise specifically from the role in blocking autophagy? Figure 4C-D speak to this somewhat, but to what extent does SNAP29 contribute to this effect?

The growth defect in in the ΔS strain arises from the lack of PR-ubiquitination. PR-ubiquitinome of the cell at 2hpi identified about 180 different substrate proteins. These belong to different GO pathways and indicate a global PR-Ub dependent rewiring of cell metabolism in response to bacterial infection (Shin et al., 2020). PR-Ub modification of ER membrane proteins cause recruitment of intracellular membranes to the bacterial vacuole which is important for the growth of the LCV and can directly affect proliferation as seen in figure 4. This recruitment is chiefly dependent on the PR-ub of STX17 but PR-Ub of other ER proteins like the ER phagy receptors may also contribute an appreciable role in the process. We see greater recruitment of the PR-Ub deficient mutant of SNAP29 to bacteria (figure 2 F and G). To test the effect of SNAP29 on bacterial replication we did the cfu assay in Legionella infected cells pretreated with SNAP29 siRNA. Depletion of SNAP29 by siRNA treatment did not affect intracellular replication of WT Legionella (possibly due to the overriding inhibitory effect of the bacterial RavZ) but increased intracellular replication of the ΔR strain. This was possibly because SNAP29 depletion by siRNA treatment protected LC3+ Legionella from lysosomal degradation (Figure S5H).

4) In the Legionella replication assays using A549 cells, are the bacteria really only dividing once or twice within the 2-day time period? Perhaps a more relevant macrophage model would provide a better measure.

We thank the reviewer for this suggestion- We repeated our cfu assay shown in extended figure 1E (comparing the replication of the different strains WT, ΔS , ΔR , $\Delta R\Delta S$) in the mouse macrophage line RAW264.7. In these cells the bacterial load is greater as these cells may be more amenable to the phagocytic uptake of Legionella-however the overall the pattern of the replication curves remained unchanged. However, these cells were difficult to transfect with siRNA and plasmids to reach

transfection/knockdown efficiencies that were comparable to our A549 cfu experiments, so we decided to not repeat the other cfu experiments in RAW264.7 cells.

5) The authors include some analysis of DupA and DupB in Figure 1, but do not build that into the model for SidE regulation of autophagy. Could they be downregulating the effects of the SidE ligases? For example, in the experiment shown in Figure 1C, in the background of the ΔR mutant, would DupA/B deletion or overexpression alter the number of acidified vesicles or LC3-positive LCVs?

The DupA and DupB proteins modulates SidE mediated PR-ubiquitination during infection. PR-Ub levels in the cell are fine-tuned to rewire the host cell metabolism. The PR-Ub levels peak at 2h in WT legionella infection, it gradually decreases at 4h and is almost undetectable at 6hpi. This sophisticated temporal regulation by Legionella was studied previously (Figure 4B from Shin et al., 2020). We do not have a strain with a deletion of DupA/B in the background of the ΔR mutant but in other infection studies we have seen that overexpression of WT DupA (but not the substrate trapping mutant DupA(H67A)) can compensate for the bacterial DuP when cells are infected with ΔDup Legionella (Figure 4B from Shin et al., 2020).

6) To what extent are the SidE family members acting redundantly in this process? Some experiments refer only to SidE, while others use SdeA. Is the ΔS Legionella strain lacking all four effectors? This is unclear from the text.

We thank the reviewer for this suggestion- we have altered the text in the introduction to clarify that SidE family members include SdeA, SdeB, SdeC and SidE. In all our infection experiments the ΔS Legionella strain lacks all 4 members of the SidE family and is therefore completely deficient of PR-Ubiquitination. In experiments without bacterial infection, SdeA transfection is used to cause PR-ubiquitination in cells or purified SdeA is used as an enzyme to catalyze in vitro PR-ubiquitination of STX17 and SNAP29.

7) The methods section requires additional details. Specifically, more information should be provided on cell culture methods, the construction of the APEX cell line, the details of all Legionella strains used in the study, quantitation of microscopy images (especially for LAMP1 staining, as well as details on the number of LCVs quantified), and the analysis of mass spectrometry data.

We have made the necessary edits and added details (highlighted in yellow) in the text.

8) The proteomics datasets should be deposited in an appropriate database and made publicly available.

The mass spectrometry proteomics data have been deposited to the ProteomeXchange Consortium via the PRIDE [1] partner repository with the dataset identifier PXD058184".

9) Some of the representative microscopy images shown are counter to what would be

expected. For example, in the WT infection shown in Figure 1B, where RavZ should be present, why are there so many LC3-positive LCVs? Also, in Figure 1E, for the ΔS mutant, there appears to be colocalization of LAMP1 and Legionella, even though the argument made in the text is that there is none.

We thank the reviewer for pointing out this discrepancy. The images in figure 1B were made from an experiment where Torin had been added to induce autophagy to all sets. We changed the images to a new experiment where Torin was not added except in the uninfected set and reanalyzed the data. For Figure 1E- the ΔS mutant does recruit LAMP1 to bacteria however the extent of LAMP1 recruitment is lesser and it does not coat the entire bacterial vacuole like what is seen in $\Delta R\Delta S$. This has been clarified in the text.

10) More analysis should be performed on the $\Delta R\Delta S$ double mutant, including quantitation of LC3-positive LCVs to show that SidE is acting downstream of this step, as well as additional EM to show fusion of these double mutant LCVs with the lysosome.

We thank the reviewer for this suggestion. We added a quantitation of LC3-positive LCVs to figure 1B. We also tested the recruitment of STX17-GFP and SNAP29-GFP to the $\Delta R\Delta S$ double mutant in an experiment which is similar to figure 2D-2F. This has been added to Extended figure 5 (panels D and E).

11) The type of ubiquitin linkage that SidE ligases catalyze is not explained clearly and categorizing it as serine ubiquitylation is very confusing for the reader. Improved clarity in the text, as well as the schematics in Figure 1A, would be very helpful. The authors should also consider distinguishing this from conventional serine ubiquitylation with an alternate nomenclature, possibly PR-ubiquitylation.

We thank the reviewer for pointing this out. We have edited the text to call it PR-ubiquitylation throughout the manuscript and have explained its difference from conventional serine ubiquitination in the introduction.

12) Can the authors justify their choice of utilizing A549 cells as a model for Legionella infection?

A549 cells is a alveolar epithelial cell line that has the naturally permissive to be infected by Legionella pneumophila. It has been shown in several studies that *L. pneumophila* is internalized and replicate effectively in A549 cells (Maruta et al., 1998; Vinzing et al., 2008; Bartfeld et al., 2009). These cells have phagocytic capability and can activate a NFkB dependent proinflammatory cytokine response in response to infection. They were used as the model system in this case because they were more amenable to transfections and are easier for microscopy-based studies compared to macrophage-based cell lines. However, we have repeated the cfu assay in RAW264.7 cells in Extended figure 1E to compare bacterial replication in A549 versus RAW264.7 cells.

13) Some panels, such as Figure 2G, are missing statistical analyses.

We added statistical analyses to figure 2G as pointed out by the reviewer.

14) Input samples should be shown for the streptavidin IP experiments presented in Figure 4G. Some sort of control showing that the STX17-APEX construct localizes to the LCV should also be included.

Input blots for indicated proteins was added to figure 4G. STX17-APEX construct has a FLAG tag- FLAG staining of STX17-APEX expressing cells infected with Legionella was added in extended figure 6B.

15) There is no clear indication of how many times certain experiments were repeated, particularly for the western blots that also lack quantitation.

We added quantification of western blots in figure 3D, 4G and 5C-F and have indicated the N for number of experiments in the figure legends.

Minor Concerns:

1) The naming of Legionella strains is not consistent throughout. Some figures use "WT", while others use "Lp02".

We have replaced all Lp02 with WT to maintain uniformity in nomenclature.

2) The magnified inset for Figure 2F is not indicated properly.

We edited it the inset now.

3) Some western blots are lacking labels, such as the top portion of Figure 2A. The experiments presented in Figures 5C and 5F are very difficult to interpret as currently labeled (some bands are not labeled, some columns are listed as "P" and others as "PR", there is no indication of what SNAP29 48- means).

We apologize for the typo. We have altered the labels and unified the nomenclature for PR-Ub as PR to maintain homogeneity.

4) The presentation of the EM images could be improved. The empty space could be used to show a representative lysosome or multivesicular body. The labels on top of the image, such as "LkL", are very difficult to read and should be replaced with arrows.

We thank the reviewer for this suggestion. We have edited the representation.

5) Different time points of infection are used for EM across WT and ΔS infections. The authors should make this consistent or otherwise rationalize their decisions.

The EM images in Figure 3E were both taken at 6 h.p.i as mentioned in the figure legend. We have rewritten in the text in the results section to avoid ambiguity.

6) The experimental schematic in Figure 4E should be altered to include the other sample comparisons studied.

We have altered the schematic as suggested by the reviewer.

Referee #2:

Mukherjee and co-workers report that Legionella SidE effectors mediate phosphor-ribosyl-linked serine ubiquitination (PR-Ub) of the autophagic SNARE proteins STX17 and SNAP29. They find that PR-Ub of STX17 increases its affinity for the ER-binding PI3K subunit ATG14, thereby promoting recruitment of ER membranes to the bacterial vacuole in a PI3K-dependent manner and the formation of bacterial replicative vacuoles. The authors also find that PR-Ub of SNAP29 prevents formation of the STX17-SNAP29-VAMP8 SNARE complex that drives fusion of autophagosomes with lysosomes, thereby preventing fusion of STX17-containing bacterial vacuoles with lysosomes. This manuscript is conceptually interesting since it indicates a novel mechanism for how a bacterium produces factors that sustain its replicative niche. Overall, the experimental data are of high quality and support the proposed mechanisms. I therefore think this manuscript would be a good fit for EMBO Journal pending successful revision.

Major point:

In a recent paper, Jian et al. reported that a SNARE complex that contains SNAP47 instead of SNAP29 mediates autophagosome-lysosome fusion in both bulk and selective autophagy (Cell Research, 2024). From Figure 4F it appears that SNAP47 is not modified by PR-Ub, and it remains unclear why SNAP47 cannot substitute for SNAP29 in vacuole-lysosome fusion in Legionella infected cells when the latter is inhibited by PR-Ub. The authors should investigate this issue.

We thank the reviewer for her/his appreciation and comments. The point about SNAP47 compensating for the loss of SNAP29 is critical and very possible in Legionella infected cells. We did not observe SNAP47 in our mass spectrometric data so it is not PR-Ub modified. We see that knockdown of SNAP29 by itself did not affect bacterial replication in a cfu assay. Under such conditions it may be possible that SNAP47 compensates for SNAP29 in the SNARE complex. The manuscript from Jian et al. is an extensive study which also shows O-GlcNAcylation of SNAP29 reduces its participation in SNARE complex formation, and under such circumstances, when SNAP29 is O-GlyNacylated, SNAP47 can replace it in the autophagosomal SNARE complex. One of the O-

GlcNacylation sites identified by Guo et al., 2014 is S61 which is in close proximity to the PR-Ub site of SNAP29 (S63), so there is a possibility of possible cross-talk between these PTMs. We think it is not directly in the scope of the current manuscript to explore the role of SNAP47 in vacuole fusion. However, we have included this possibility in the discussion section and have cited the paper by Jian et al, 2024.

Dear Ivan,

We have now received re-review reports from both referees, which I have included below. As you will see, whilst Referee #2 is satisfied with your revisions, Referee #1 maintains that the mechanistic depth of your work needs to be improved. After carefully going through your point-by-point response and manuscript text, I conclude that questions on the roles of PR-deubiquitination and SNAP47 function would most realistically be addressed in follow-up studies. However, I remain curious about how PR-ubiquitination and phosphorylation of STX17 at Ser202 can both define functionally distinct proteins pools on one hand (as stated in your discussion), and have similar effects on the other (as stated in your point-by-point response). I would like you to address and clarify this point.

In addition, there are some remaining editorial points which need to be addressed. In this regard would you please:

- label the corresponding author in the manuscript file and include your email address here,
- acknowledge the following funding in our online submission system: an Alexander von Humboldt Stiftung postdoctoral fellowship; LYSOFOR2625; DFG-LYSOFOR2625 and the Dutch Research Council (NWO), project NEMI, number 184.034.014; innovation programme under the Marie Skłodowska-Curie grant agreement No 765445,
- in the reference section, use the format of 10 authors + et al. for longer author lists,
- rename the conflict of interests statement as a "Disclosure and competing interests statement",
- remove the AC/CrediT section from the text,
- remove the reference to data not shown on page 15 of the manuscript,
- check potential typos referring to Fig. 5 and Fig. 6 in the manuscript text,
- correct author checklist so that pink boxes (column E) are blank if the response in column D is 'Not Applicable',
- supply a link to the proteomic raw data (made publicly available) that are summarised in Figure 4F and update the author checklist,
- state exact p values in the legends of figures 1B-E; 2E, G; 3B, D, E; 4A, B, C, D, G; 5C, D, E, F; EV1 C-F; EV5 D, E, G, H,
- indicate the statistical test used for data analysis in the legends of figures 4F, EV5 A, EV6 D, E,
- define */ **/ ***/ **** in the legend of figure EV5 B. If p values, please indicate the statistical test used, where appropriate specify whether it was one-sided or two-sided, and whether adjustments were made for multiple comparisons,
- define n in the legends of figures 4F, EV6 D, E,
- define the error bars in the legends of figures 1B, 3D, E; 4A,G; 5C-F; EV5 A, B,
- define the measure of centre for the error bars in the legend of figures EV1 C,
- include a scale bar in figure 2E,
- define the scale bar in figures 1B, EV5 F,
- include and define a scale bar in figures 4A, B; EV1 D; EV5 D, E; EV6 B,
- define the dotted lines in the legend of figures 1C, E; 2D, F; 3A, 4A, B; EV1 D, EV5 B, D, E, F,
- define the white arrows in the legend of figures EV6 B,
- rename "Summary" as the "Abstract", and
- correct the section order as follows: Title page - Abstract & Keywords - Introduction - Results - Discussion - Methods - Data Availability - Acknowledgements - Disclosure and Competing Interests Statement - References - Figure Legends - Table(s) - Expanded View Figure Legends.

We include a synopsis of the paper (see <http://emboj.embopress.org/>). Please provide me with a general summary image, a two sentence statement and 3-5 bullet points that capture the key findings of the paper.

I am looking forward to receiving your revised manuscript.

EMBO Press is an editorially independent publishing platform for the development of EMBO scientific publications.

Best wishes,

William

William Teale, PhD
Editor
The EMBO Journal
w.teale@embojournal.org

We realize that it is difficult to revise to a specific deadline. In the interest of protecting the conceptual advance provided by the work, we recommend a revision within 3 months (9th Jun 2025). Please discuss the revision progress ahead of this time with the editor if you require more time to complete the revisions. Use the link below to submit your revision:

Referee #1:

Mukherjee and Bhattacharya et al have enhanced the presentation of their work in the resubmitted manuscript and appear to have performed additional analyses of existing data. The authors do indicate that input blots were included into figure 4G to aid in the interpretation of STX17 interactions, but these appear to be missing. Very little has been done to address the major experimental concerns raised by both reviewers, namely in the impact of PR-ubiquitylation on STX17 and SNAP29 function, the role of DupA/DupB regulation, the relevance for Legionella replication, and the interplay with SNAP47. This leaves very little room for mechanistic interpretation of how PR-ubiquitylation is influencing STX17, SNAP29, and possibly also SNAP47 activity in support of Legionella infection.

Referee #2:

The authors have successfully addressed my comments; I recommend publication without further review.

Dear Ivan,

We have now received re-review reports from both referees, which I have included below. As you will see, whilst Referee #2 is satisfied with your revisions, Referee #1 maintains that the mechanistic depth of your work needs to be improved. After carefully going through your point-by-point response and manuscript text, I conclude that questions on the roles of PR-deubiquitination and SNAP47 function would most realistically be addressed in follow-up studies. However, I remain curious about how PR-ubiquitination and phosphorylation of STX17 at Ser202 can both define functionally distinct proteins pools on one hand (as stated in your discussion), and have similar effects on the other (as stated in your point-by-point response). I would like you to address and clarify this point.

To better address this question, we have rewritten the part of the discussion to increase clarity:

“During the initiation of autophagy, two major protein complexes are recruited to STX17-positive phagophore assembly sites: the PI(3)-kinase complex—which includes Vps34, Vps15, ATG14L, and Beclin1—and the ULK1 complex, composed of ATG101, ULK1/2, FIP200, and ATG13. Under autophagy-inducing conditions, the cellular kinase TBK1 phosphorylates STX17 at serine 202, a modification that promotes the recruitment of ATG13 and FIP200, facilitating the formation of the mammalian pre-autophagosomal structure (mPAS).

Interestingly, during *Legionella* infection, the pathogen exploits the same serine residue (S202) for phosphoribosyl-linked ubiquitination (PR-Ub). This modification enhances STX17's interaction with ATG14L, a component of the PI3K complex. Although phosphorylation and PR-ubiquitylation represent distinct post-translational modifications with potentially different impacts on STX17's interaction landscape, both ultimately converge on a similar functional outcome: promoting the formation of autophagosomes during canonical autophagy, or autophagosome-like bacterial vacuoles during *Legionella* infection.

In addition, there are some remaining editorial points which need to be addressed. In this regard would you please:

- label the corresponding author in the manuscript file and include your email address here
Done

- acknowledge the following funding in our online submission system:
an Alexander von Humboldt Stiftung postdoctoral fellowship; LYSOFOR2625; DFG-LYSOFOR2625 and the Dutch Research Council (NWO), project NEMI, number 184.034.014; innovation programme under the Marie Skłodowska-Curie grant agreement No 765445,

- in the reference section, use the format of 10 authors + et al. for longer author lists
Reference section was reformatted as per the EMBO format

- rename the conflict of interests statement as a "Disclosure and competing interests statement",
Done.

- remove the AC/Credit section from the text,

Done

- remove the reference to data not shown on page 15 of the manuscript,

Done. We rewrote this portion excluding any reference to data not mentioned in the manuscript.

- check potential typos referring to Fig. 5 and Fig. 6 in the manuscript text

We apologize for this discrepancy and have altered this in the manuscript.

- correct author checklist so that pink boxes (column E) are blank if the response in column D is 'Not Applicable',

Done

- supply a link to the proteomic raw data (made publicly available) that are summarised in Figure 4F and update the author checklist,

Done

- state exact p values in the legends of figures 1B-E; 2E, G; 3B, D, E; 4A, B, C, D, G; 5C, D, E, F; EV1 C-F; EV5 D, E, G, H,

Done

- indicate the statistical test used for data analysis in the legends of figures 4F, EV5 A, EV6 D, E

Done,

- define */ **/ ***/ **** in the legend of figure EV5 B. If p values, please indicate the statistical test used, where appropriate specify whether it was one-sided or two-sided, and whether adjustments were made for multiple comparisons,

Done

- define n in the legends of figures 4F, EV6 D, E,

Done

- define the error bars in the legends of figures 1B, 3D, E; 4A,G; 5C-F; EV5 A, B,

Done

- define the measure of centre for the error bars in the legend of figures EV1 C,

The error bars are +/-SD

- include a scale bar in figure 2E,

- define the scale bar in figures 1B, EV5 F,

- include and define a scale bar in figures 4A, B; EV1 D; EV5 D, E; EV6 B,

- define the dotted lines in the legend of figures 1C, E; 2D, F; 3A, 4A, B; EV1 D, EV5 B, D, E, F,

- define the white arrows in the legend of figures EV6 B,

- rename "Summary" as the "Abstract",

Done

- correct the section order as follows: Title page - Abstract & Keywords - Introduction - Results -

Discussion - Methods - Data Availability - Acknowledgements - Disclosure and Competing Interests Statement - References - Figure Legends - Table(s) - Expanded View Figure Legends.

Done

We include a synopsis of the paper (see <http://emboj.embopress.org/>). Please provide me with a

general summary image, a two-sentence statement and 3-5 bullet points that capture the key findings of the paper.

Done

Referee #1:

Mukherjee and Bhattacharya et al have enhanced the presentation of their work in the resubmitted manuscript and appear to have performed additional analyses of existing data. The authors do indicate that input blots were included into figure 4G to aid in the interpretation of STX17 interactions, but these appear to be missing. Very little has been done to address the major experimental concerns raised by both reviewers, namely in the impact of PR-ubiquitylation on STX17 and SNAP29 function, the role of DupA/DupB regulation, the relevance for Legionella replication, and the interplay with SNAP47. This leaves very little room for mechanistic interpretation of how PR-ubiquitylation is influencing STX17, SNAP29, and possibly also SNAP47 activity in support of Legionella infection.

We acknowledge that the mechanistic insights the reviewer refers to are not fully explored within the current scope of our study. These questions - regarding the precise impact of PR-ubiquitylation on STX17 and SNAP29 function, the regulatory role of DupA/DupB, the relevance for Legionella replication, and the potential interplay with SNAP47 - are indeed highly significant. However, they require comprehensive, in-depth investigation that extends beyond the scope of the present manuscript.

Our use of $\Delta DupA/DupB$ Legionella aims to stabilize PR-ubiquitylation events by limiting their removal, thereby creating a cellular environment in which PR-ubiquitylated targets can be more readily identified. However, to dissect the cellular phenotypes modulated by PR-ubiquitylation under more physiological conditions, we believe that comparing WT bacteria to ΔS mutants offers more relevance. WT *Legionella* exhibit dynamic PR-Ub modification, with a peak occurring around 1-2 hours post-infection and a marked decrease by 4 hours post-infection. Future studies comparing WT and $\Delta DupA/DupB$ infections will be essential for resolving the temporal dynamics and downstream consequences of PR-ubiquitylation during infection.

We have addressed the relevance of PR-ubiquitylation of STX17 and SNAP29 to bacterial replication in Figures 4C, 4D, and E5H. Specifically, we show that PR-ubiquitylation of STX17 supports *Legionella* replication (Figures 4C, 4D). While PR-ubiquitylation of SNAP29 did not significantly affect replication in WT bacteria—likely due to the overriding effect of RavZ—in its absence (in ΔR and $\Delta R\Delta S$ strains), SNAP29 knockdown promotes infection (Figure E5H).

As for the potential functional interplay between SNAP47 and SNAP29, we agree that this is a compelling direction for future work. However, it falls outside the immediate scope of this study and will be explored in detail in a follow-up project currently underway in our lab.

Referee #2:

The authors have successfully addressed my comments; I recommend publication without further review.

We thank the reviewer for his/her comments which have been very valuable to improve this manuscript.

Dear Ivan,

I am pleased to inform you that your manuscript has been accepted for publication in the EMBO Journal.

Congratulations to you and to all involved!

Yours sincerely,

William

William Teale, PhD
Editor
The EMBO Journal
w.teale@embojournal.org
